# Parametric Copula-GP model for analyzing multidimensional neuronal and behavioral relationships

Nina Kudryashova[1]*, Theoklitos Amvrosiadis[2], Nathalie Dupuy[2], Nathalie Rochefort[2,3], Arno Onken[1]

**1** School of Informatics, University of Edinburgh, Edinburgh, United Kingdom, **2** Centre for Discovery Brain Sciences, University of Edinburgh, Edinburgh, United Kingdom, **3** Simons Initiative for the Developing Brain, University of Edinburgh, Edinburgh, United Kingdom

* nkudryas@inf.ed.ac.uk

**Data Availability Statement:** The data and the source code used to produce the results and analyses presented in this manuscript are available on Github: https://github.com/NinelK/CopulaGP.

## Abstract

One of the main goals of current systems neuroscience is to understand how neuronal populations integrate sensory information to inform behavior. However, estimating stimulus or behavioral information that is encoded in high-dimensional neuronal populations is challenging. We propose a method based on parametric copulas which allows modeling joint distributions of neuronal and behavioral variables characterized by different statistics and timescales. To account for temporal or spatial changes in dependencies between variables, we model varying copula parameters by means of Gaussian Processes (GP). We validate the resulting Copula-GP framework on synthetic data and on neuronal and behavioral recordings obtained in awake mice. We show that the use of a parametric description of the high-dimensional dependence structure in our method provides better accuracy in mutual information estimation in higher dimensions compared to other non-parametric methods. Moreover, by quantifying the redundancy between neuronal and behavioral variables, our model exposed the location of the reward zone in an unsupervised manner (i.e., without using any explicit cues about the task structure). These results demonstrate that the Copula-GP framework is particularly useful for the analysis of complex multidimensional relationships between neuronal, sensory and behavioral variables.

## Author summary

Understanding the relationship between a set of variables is a common problem in many fields, such as weather forecast or stock market data. In neuroscience, one of the main challenges is to characterize the dependencies between neuronal activity, sensory stimuli and behavioral outputs. A method of choice for modeling such statistical dependencies is based on copulas, which disentangle dependencies from single variable statistics. To account for changes in dependencies, we model changes in copula parameters by means of Gaussian Processes, conditioned on a task-related variable. The novelty of our approach includes 1) explicit modeling of the dependencies; and 2) combining different copulas to

**Funding:** This work was supported by the Engineering and Physical Sciences Research Council (grant [EP/S005692/1], to A.O.), the Precision Medicine Doctoral Training Programme (Medical Research Council grant number [MR/N013166/1], to T.A.), the Wellcome Trust and the Royal Society (Sir Henry Dale fellowship to N.L.R. [102857/Z/13/Z]), the RS MacDonald Charitable Trust Seedcorn Grant (PWC ref.29 to N.L.R.), the Simons Initiative for the Developing Brain (to N.L. R.), and the Biotechnology and Biological Sciences Research Council (BBSRC) (Responsive Mode Research Grant [BB/T007907/1] to N.L.R.). This project has received funding from the European Research Council (ERC) under the European Union's Horizon 2020 research and innovation program (grant agreement 866386). The funders had no role in study design, data collection and analysis, decision to publish, or preparation of the manuscript.

**Competing interests:** The authors have declared that no competing interests exist.

describe experimentally observed variability. We validate the goodness-of-fit as well as information estimates on synthetic data and on recordings from the visual cortex of mice performing a behavioral task. Our parametric model demonstrates significantly better performance in describing high dimensional dependencies compared to other commonly used techniques. We demonstrate that our model can estimate information and predict behaviorally-relevant parameters of the task without providing any explicit cues to the model. Our results indicate that our model is interpretable in the context of neuroscience applications, scalable to large datasets and suitable for accurate statistical modeling and information estimation.

This is a *PLOS Computational Biology* Methods paper.

## Introduction

Recent advances in imaging and recording techniques have enabled monitoring the activity of hundreds to thousands of neurons simultaneously [1–3]. These recordings can be made in awake animals engaged in experimentally-constrained behavioral tasks or natural behaviors [4–6], which further augments these already large datasets with a variety of behavioral variables. These complex high dimensional datasets necessitate the development of novel analytical approaches [7–10] to address two central questions of systems and behavioral neuroscience: how do populations of neurons encode information? And how does this neuronal activity relate to the observed behavior? In machine learning terms, both of these questions translate into understanding the high-dimensional multivariate dependencies between the recorded variables [4, 11–14, 14].

The experimentally recorded neuronal and behavioral variables may operate at different timescales and exhibit different statistics. While neuronal spiking occurs on a temporal scale of milliseconds [1–3], the behavioral variables span timescales from milliseconds to hours and even days [5, 6, 15, 16]. The overlap of these multi-scale dynamics results in a complex joined distribution of the simultaneously recorded activity. Therefore, the newly proposed analytical tools must be able to accommodate such complex statistics and uncover the underlying dependencies.

A method of choice for modeling statistical dependencies between variables with drastically different statistics is based on *copulas*, which separate marginal (i.e. single variable) statistics from the dependence structure [17]. For this reason, copula models are particularly effective for *mutual information* estimation [18, 19], which quantifies how much knowing one variable reduces the uncertainty about another variable [20]. Copula models can also escape the 'curse of dimensionality' by factorising the multi-dimensional dependence into pair-copula constructions called *vines* [21, 22]. This factorization of the dependence in vine copulas also, to a limited extent, accounts for the higher-order correlations between the variables [23].

Copula models have been applied to spiking activity [24–27] in cortical areas, spiking activity with local-field potential [23] and somatic calcium transients [28]. However, these models assumed that the dependence between variables was static, and could not capture neither the dynamics nor the changes with respect to continuous (non-discrete) external variables [12, 29]. In this paper, we demonstrate the limitations of this static copula approach using simple

toy models with time-dependent stimuli and dynamically coupled neurons. We demonstrate that, in order to analyse the underlying dynamic computation, it is critical to explicitly model the continuous time- or context-dependent changes in the relationships between neuronal and behavioral variables.

Here, we use a copula-based approach with explicit conditional dependence for the parameters of the copula model and approximate these latent dependencies with Gaussian Processes (GP). It was previously shown that such a combination of parametric copula models with GP priors outperforms static copula models [30] and even dynamic copula models on many real-world datasets, including weather forecasts, geological data or stock market data [31]. Yet, this Copula-GP approach has never been applied to neuronal recordings before.

Here, we propose and validate a method that is interpretable in the context of neuroscience applications, scalable to large datasets and suitable for accurate probability density and information estimation. We first briefly introduce the Copula-GP model, demonstrating its utility for a simple neuroscience example with two dynamically coupled neurons and interpreting each component of the model. We then validate our model on synthetic data and compare its performance against other commonly used information estimators, and show that it has better scaling to higher dimensions. Next, we demonstrate the utility of the method on real neuronal and behavioral data, acquired with two-photon calcium imaging in the primary visual cortex of awake behaving mice performing a rewarded task [6, 32]. We demonstrate that our model can estimate mutual information and predict behaviorally-relevant parameters of the task without providing any explicit cues to the model. Finally, we validate the model with a high-dimensional dataset with hundreds neurons and multiple behavioral variables. These results demonstrate that the Copula-GP framework is particularly useful for the analysis of complex multidimensional relationships between neuronal, sensory and behavioral variables.

## Results

### 1 Copula-GP model

Many questions in neuroscience require accurate statistical models of neuronal responses to a certain stimulus. For instance, such models can be used for Bayesian decoding of the stimulus [20] or for information-theoretic analysis [33] in order to assess the coding precision of a neuronal population. Here we describe a general framework for constructing such statistical models, called Copula-GP; the technical details related to model implementation and fitting can be found in Methods.

Our statistical model is based on **copulas**: multivariate distributions with uniform marginals. Sklar's theorem [34] states that any multivariate joint distribution can be written in terms of univariate marginal distribution functions $p_i(y_i)$ and a unique copula which characterizes the dependence structure:

$$p(y_1, \ldots, y_N) = c(F_1(y_1) \ldots F_N(y_N)) \times \prod_{i=1}^{N} p_i(y_i). \tag{1}$$

Here, $F_i(\cdot)$ are the marginal cumulative distribution functions (CDF) corresponding to the probability density functions (PDF): $p_i(y_i) = dF_i(y)/dy|_{y=y_i}$.

The model in (1) has been previously applied to neuronal data [18, 23, 35]. In this paper, we extend that model by conditioning it on a continuous variable $x$:

$$p(\mathbf{y}|x) = c(F_1(y_1|x), \ldots, F_N(y_N|x)|x) \times \left[ \prod_{i=1}^{N} p_i(y_i|x) \right]. \tag{2}$$

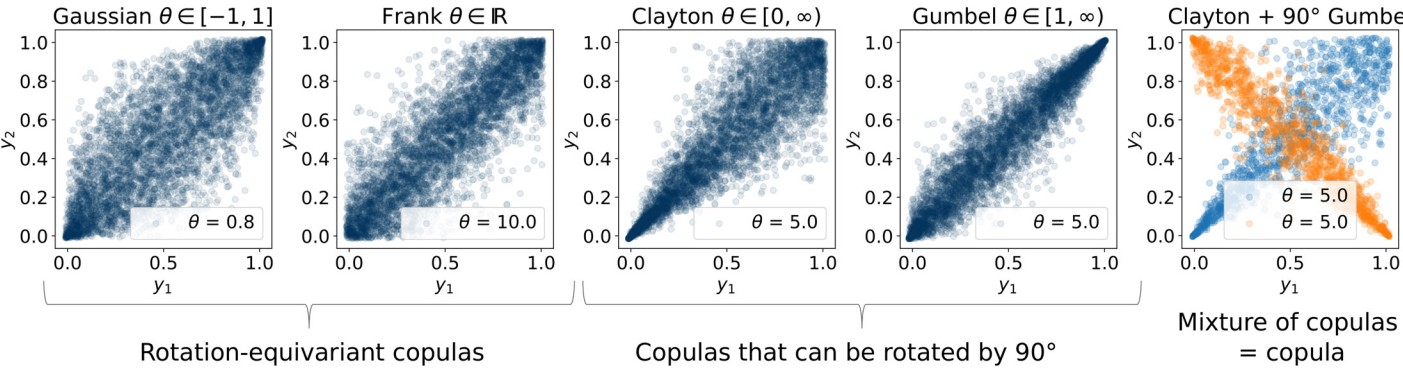

**Fig 1. Copula families used in the mixture models in our framework.** Each panel shows a scatter plot of samples drawn from a parametric copula family (named in the title of each plot) with a fixed parameter $\theta$ (shown in the bottom right corner). In total, we used 10 different copula elements—Gaussian + Frank + 4 × Clayton + 4 × Gumbel—to construct copula mixtures. The rightmost figure shows a mixture of two copulas from different copula families taken with equal mixing weights (0.5). Blue points here correspond to the samples drawn from a Clayton copula, orange points—to a 90˚-rotated Gumbel copula. Note, that a mixture of copulas is a copula itself.

where $x$, depending on the design of the experimental task, can represent any continuous task-related variable such as time, phase, coordinate in space, velocity, direction or orientation. In the further sections we will show that if this variable $x$ also uniquely determines the stimulus, then the components of the model (2) can also be interpreted in terms of 'noise correlations' (i.e. joint trial-to-trial variability of responses to a given stimulus).

We are using Gaussian Processes (GP) to model the conditional dependencies of copula parameters (i.e. $c(.|x)$), following the approach by Hernández-Lobato et al. [31] which was originally developed for analysis of financial time-series. GPs are ideally suited for copula parametrization, as they make the dependencies explicit and provide an estimate of the uncertainty in model parameters, which we utilize in our model selection process (see Sec. 6 in Methods). Similarly to previous computational studies [23, 28, 30, 36, 37], we use vine copula constructions to scale the method to higher dimensions (Sec. 7). Here, we extend the Copula-GP framework with more flexible copula mixture models (see Fig 1 and Methods), suitable for inter-neuronal dependencies, and utilise the repeated trial structure in order to estimate conditional marginal distributions $p_i(y_i|x)$ (e.g. distributions of single neuron responses at every moment $x$ in the trial). We develop model selection algorithms, based on the fully-Bayesian Watanabe–Akaike information criterion (WAIC), that construct the best suited mixture models for a given data set. In the following section we show the utility of our extended Copula-GP model for analysis of neuronal and behavioral data.

## 2 Copula-GP separates noise correlations from stimulus statistics

We first examine a synthetic 'toy' model of calcium imaging data in order to illustrate and interpret the components of our Copula-GP model. We use a Generalized linear model (GLM) [38] and a model of the kinetics of the calcium indicator [39] in order to generate synthetic calcium imaging data of the activity of two neurons (see Methods). GLMs are widely used to model population spike trains based on spike history filters and coupling filters. While these models can be readily fit to small neuronal populations, they have parameter identifiability issues that lead to degraded performance in predicting neuronal activity patterns [40]. GLMs are also not directly applicable to calcium imaging data and require some additional spike inference algorithm (deconvolution, e.g. Deneux et al. 2016 [39]) to obtain spikes. So, instead of fitting GLMs, we propose to explicitly model the dynamic neuronal activity patterns

(including the calcium dynamics) with Copula-GP. Our approach is complementary to GLMs: while GLMs consider time-lagged interactions via temporal spike history filters and cannot model instantaneous coactivation [41], our model focuses exclusively on the instantaneous correlations between calcium signals.

Calcium imaging in neurons is used as a proxy for monitoring their spiking activity since action potentials are tightly coupled to large increases in intracellular free calcium concentration [42]. Two-photon imaging is used to monitor fluorescence changes of calcium indicators in large population of neurons, with single cell resolution. When dense populations of neurons are imaged, the signals imaged in a given focal plane can be contaminated by signals from a volume around this plane. Different methods for correcting such out-of-focus contamination have been developed [43]. We assumed that each decontaminated recording $y_i(t)$ of calcium dynamics for each neuron is a function of its spiking activity $n_i(t)$, which is independent of the activity of other neurons (i.e. $y_i(n_1 \ldots n_N, t) = y_i(n_i, t)$, given time). This assumption is also implicitly made in deconvolution algorithms, which incorporate inductive biases for calcium dynamics and are also applied to individual neurons independently. We will consider the decontaminated recordings $y_i(t)$ to be a surrogate of neural spike trains $n_i(t)$, which can be approximated by blurring (convolving) the spikes with an exponential kernel (Deneux et al. 2016 [39]). The time constant in this kernel corresponds to calcium decay and also sets the time scale of interactions between spiking neurons, which would be captured by an instantaneous correlation of their calcium traces. Keeping these assumptions in mind, we aim to build instantaneous dependence models of calcium recordings for statistical and information-theoretic analysis.

We first consider a pair of neurons, which have the same tuning (receive the same input $x(t)$ in Fig 2A), but are **not** coupled. Since the spiking activity of these neurons in response to a given stimulus is completely independent, we expect that their calcium traces, which are a temporally blurred versions of spike trains, to also be independent given the stimulus. A few simulated calcium transients (fluorescence across time) for one of these identical neurons are shown in Fig 2B.

By design, the activity of these neurons is independent given the stimulus (Fig 2A), but they do respond similarly to the stimulus. As a result of a naive analysis of their joint statistics, such as measuring correlation between recorded activities over time, their activity would appear dependent. The structure of this dependence can be visualized with a static copula model, as in (1).

To fit a copula-based model to a given dataset, one typically starts with the marginals (i.e. single variable distributions). Here, we model the single neuron marginals with their empirical distributions, estimated directly from the data in the form of empirical cumulative distribution functions (eCDFs). We then use these eCDFs to project the simulated neuronal recordings onto a unit cube using the probability integral transform: $F(y_i) \to u_i \sim U_{[0,1]}$, such that each $u_i$ becomes uniformly distributed. Note, that both the neural responses $y_i$ and their transformed versions $u_i$ still depend on both the stimulus $x(t)$ and time $t$. We can confirm that by visualizing the empirical copula $c(u_1 \ldots u_N) = c(F_1(y_1) \ldots F_N(y_N))$ (Fig 2C). The colors of the data points here indicate time $t$. Similarly colored clusters of points in copula space (e.g. green in Fig 2C) indicate that both the dependence and the marginals in such an unconditional model depend on time $t$ and on the presented stimulus $x(t)$. Hence, such dependence characterizes the joint statistics of neural responses to stimulus $x(t)$, but not the interaction between neurons.

Spurious correlations between neuronal recordings are a well known problem when modeling responses to complex stimuli (e.g. time-dependent sequences of stimuli $x(t)$), which can be solved by distinguishing between **noise** and **stimulus** correlations [40, 44]. In our Copula-GP

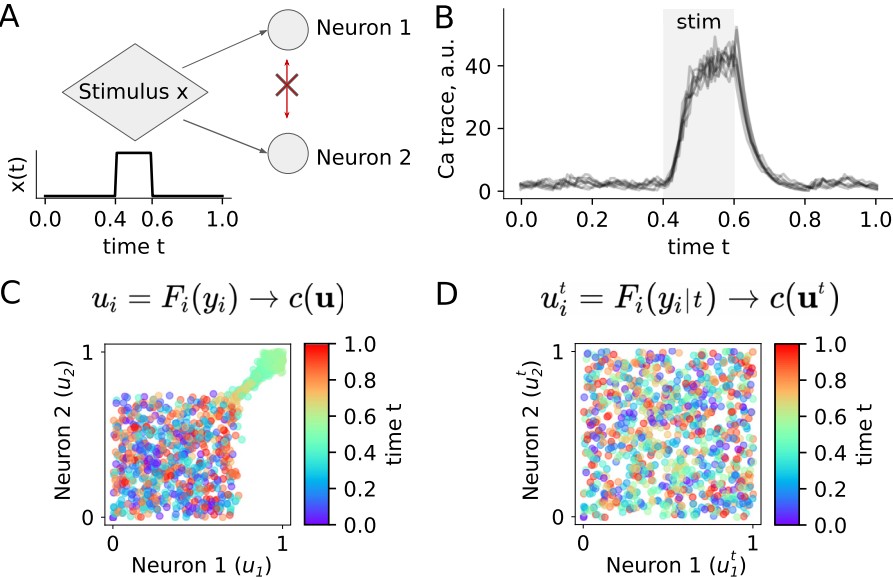

**Fig 2. Copula-GP finds that uncoupled neurons are independent given the stimulus. A** GLM model of two identical uncoupled neurons that receive the same time-dependent input $x(t)$; **B** simulated calcium transients (fluorescence across time) showing dynamic responses to the stimulus $x(t)$ for one of the neurons; **C** calcium transients of two neurons ($y_1(t), y_2(t)$) projected onto a unit cube by the probability integral transform based on unconditional marginals; colored points show transformed samples ($u_1, u_2$) corresponding to times $t$ (color-coded). The clusters of similarly colored points (e.g. green) illustrate that the copula $c(\mathbf{u})$ depends on time $t$; the particular shape and the location of the clusters depends on the function $x(t)$; only 10% of data-points are shown (selected randomly). **D** same as C, but based on conditional marginals $F_i(y_i|t)$. The resulting empirical copula describes 'noise correlations' between two neurons. The colored data-points ($u_1^t, u_2^t$) are uniformly distributed on the unit square, which suggests that there is no noise correlation between these neurons, the copula $c(\mathbf{u}^t)$ is independent of time $t$, and the neurons are independent given the time-dependent stimulus.

model, this separation is achieved by using **conditional marginals and copulas**:

$$p(\mathbf{y}|x(t), t) = p(\mathbf{y}|t) = \underbrace{c(F_1(y_1|t), \ldots, F_N(y_N|t)|t)}_{\text{noise correlations}} \times \underbrace{\left[\prod_{i=1}^{N} p_i(y_i|t)\right]}_{\text{stimulus correlations}}.$$

Here, we only assume that the stimulus $x(t)$ is fully determined by time $t$. In general, our framework (2) is applicable to any complex stimuli determined by any continuous variable (e.g. time, phase, coordinate in space, velocity, direction, orientation, etc.). The conditional marginals $p_i(y_i|t)$ in this model account for the **within-trial** variability (e.g. dynamics of responses), while the copula model accounts for the **trial-to-trial** variability in neural responses.

The conditional marginals $p_i(y_i|t)$ are estimated using the non-parametric fastKDE [45] algorithm. This approach is suitable for relatively large datasets, which have enough data points for direct estimation of the eCDF (see Sec D in S3 Text). If the number of datapoints is insufficient, one might consider using parametric marginal models instead. Note that the fastKDE also assumes smooth changes in the marginal distributions with respect to the conditioning variable. The corresponding conditional marginal CDFs $F_i(y_i|t)$ are then used to map the data onto a unit hypercube using: $F_i(y_i|t) \rightarrow u_i^t \sim U_{[0,1]}$, such that $u_i^t$ is uniformly distributed *for any* $t$. The resulting empirical copula correctly demonstrates the absence of dependence between $y_1(t)$ and $y_2(t)$, as the points are uniformly distributed on a unit square. The

density of points in Fig 2D illustrates that for every given value of the transformed neural response $u_1^t$, the conditional distribution of the other neural responses is the same $(p(u_2^t|u_1^t) = p(u_2^t) \sim U_{[0,1]})$. This means that the variables $u_1^t$ and $u_2^t$ are independent $p(u_1^t, u_2^t) = p(u_2^t|u_1^t)p(u_1^t) = p(u_1^t)p(u_2^t)$, and their copula is the Independence copula. This result demonstrates that once we made the distinction between noise correlations and stimulus correlations, we could correctly identify that these two neurons are uncoupled (Fig 2A).

The noise correlations were previously linked to anatomical and functional relationships between cortical neurons [46, 47]. However, these relationships should be interpreted with caution, since correlation between neurons does not necessarily imply the mechanistic coupling between them [48]. Despite lack of mechanistic interpretability, accurate modelling of noise correlations is useful for understanding neural code from the information-theoretic perspective. Generally, noise correlations themselves can depend on the stimulus [49], and taking their stimulus dependence into account can improve the decoding accuracy [44, 50, 51]. In order to model such tuning of the noise correlations, not only the marginals but also the corresponding copula $c(\ldots|t)$ itself must be conditioned on $t$, as in (2).

We next consider two **coupled** neurons: one excitatory and one inhibitory (Fig 3A.i). They again receive the same input $x(t)$ as in Fig 2A, in all trials. We added two time-dependent filters $h_{12}$ and $h_{21}$, that couple the spike train history of each neuron to the other (Fig 3A.ii) [38]. The synthetic calcium traces (Fig 3B and 3C) demonstrate some non-trivial activity in both neurons after the stimulus presentation window, where all of the recurrent circuit dynamics unfolds. We expect that these dynamics will be reflected in some non-independent trial-to-trial co-variability of neural responses. Since one neuron inhibits the other, we also expect that their responses will be overall negatively correlated. While the unconditional copula analysis again demonstrates strong stimulus correlations (Fig 3D), the conditional copula (conditioned on the stimulus) reveals some structure in noise correlations (Fig 3E, see more points are concentrated in the upper-left corner).

In order to analyse the structure of the noise correlations in this example, we apply our Copula-GP model. For simplicity, here we use a single 90˚-rotated Clayton copula, parameterized by a Gaussian Process (GP); for details regarding the inference scheme, see Methods. The inferred GP parameter reconstructs the stimulus-dependent changes in noise correlations (Fig 3F), which are most pronounced after the stimulation window. The corresponding Clayton copula model can accurately describe the shape of the conditional dependence, which we quantify with the proportion of variance explained $\overline{R^2}$ in Fig 3G (see Sec. 10 in Methods).

This noise correlation model in Fig 3G can be interpreted as follows. The red point in the middle of each plot in Fig 3G corresponds to the median response for both neurons: 50% of recorded responses from a given neuron were higher, while 50% were lower. More generally, the values of the transformed neural responses $u^t$ correspond to the percentile scores of each response $y$ in a marginal distribution of neural responses. The shades in Fig 3G correspond to the probability density, i.e. how likely it is to jointly observe a pair of responses $u_{ex}^t$ and $u_{inh}^t$. Typically, most of the density is concentrated along one of the diagonals of the unit square: around $u_{ex}^t = u_{inh}^t$ if neurons are positively correlated, and around $u_{ex}^t = -u_{inh}^t$ if neurons are negatively correlated. In this case, the density is mostly concentrated around the negative diagonal, suggesting that the responses of these two neurons are generally negatively correlated (as expected, given that one of the neurons inhibits the activity of the other). However, this dependence also has a heavy tail: a high degree of association between some of the extreme values of the variables. We can observe this association as a probability mass concentrated in the corner of the unit square (see orange circles in Fig 3G). It indicates that there is a high probability to observe a strongly activated inhibitory neuron together with the inactivated excitatory neuron.

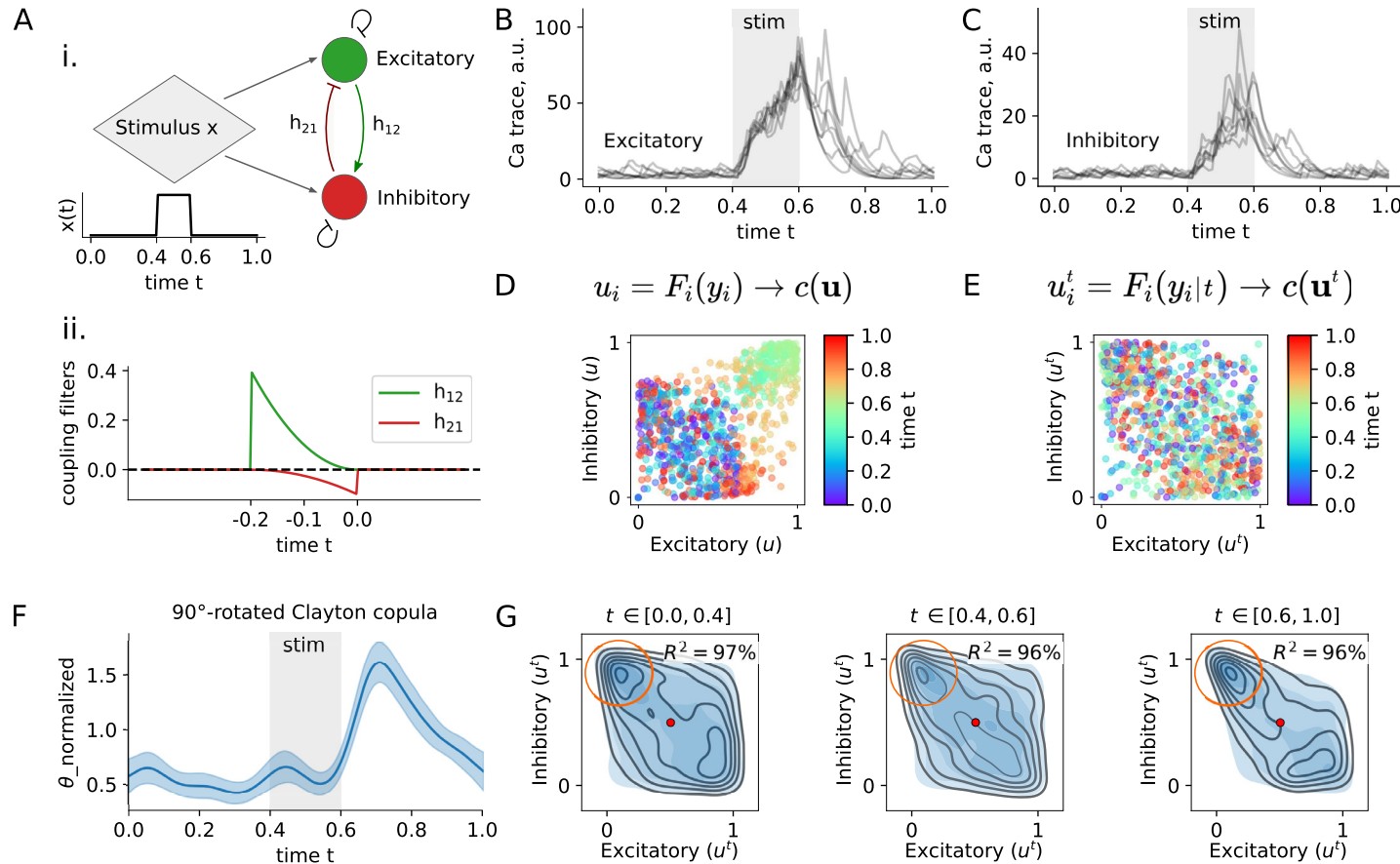

**Fig 3. Copula-GP describes the noise correlations between dynamically coupled neurons with a Clayton copula. A** i. GLM model of two coupled neurons (excitatory and inhibitory) that receive the same time-dependent input $x(t)$; ii. the spike history coupling filters $h_{12}$ and $h_{21}$; **B-C** simulated calcium transients (fluorescence across time) showing dynamic responses to the stimulus $x(t)$ for excitatory and inhibitory neurons, respectively; **D** calcium transients of two neurons $(y_1(t), y_2(t))$ projected onto a unit cube by the probability integral transform based on unconditional marginals; colored points show transformed samples $(u_1, u_2)$ corresponding to times $t$ (color-coded). The clusters of similarly colored points (e.g. green) illustrate that the copula $c(\mathbf{u})$ depends on time $t$; the particular shape and the location of the clusters depends on the function $x(t)$. **E** same as D, but based on conditional marginals $F_i(y_i|t)$. The resulting copula describes 'noise correlations' between two neurons. The colored data-points $(u_1^t, u_2^t)$ are **not** uniformly distributed on the unit square, which suggests that the noise correlation between these neurons and the copula $c(\mathbf{u}^t)$ itself depends on time $t$. **F** Clayton copula parameter ($\theta$) that characterizes the strength of the non-linear noise correlation between neurons (see Methods for details); **G** probability density plots illustrating the stimulus-dependent shape of noise correlations. The empirical dependence estimated from data samples is shown with black outlines, while the predictions of the Clayton copula model are shown in shades of blue. The proportion of the variance explained $\overline{R^2}$ is indicated in the upper-right corner for each time interval. Orange circles indicate the heavy tail of the distribution, which can be best seen in the range $t \in [0.6, 1.0]$ where the variables are stronger correlated.

Such asymmetry in the joint trial-to-trial variability of two neurons suggests that their dependence is non-linear: their dependence is not well characterized by a single linear correlation coefficient as there is additional structure in the dependence beyond linear association. In our GLM model, this non-linearity stems from the different timescales of spike history filters: fast inhibitory and slow excitatory neurons.

The confidence intervals (95% CI, two standard deviations) shown in Fig 3F correspond to the uncertainty in model parameters, captured by the Gaussian process. The uncertainty in model parameters is not essential for this example, but necessary when modeling high-dimensional datasets with relatively low sample numbers, which is often the case in neuronal data with hundreds to thousands of recorded neurons and only few hundreds of trials. These uncertainties can be propagated through the model and produce the uncertainty measures for the

decoded stimulus (when using Copula-GP for Bayesian decoding) or in estimated mutual information.

As we demonstrated in these synthetic examples, our Copula-GP model can separate the statistics corresponding to the network from the single unit statistics, or, in other words, distinguish (stimulus-dependent) noise correlations (i.e. shared neuronal trial-to-trial variability) from stimulus correlations. Moreover, it provides a visualization of the shape of noise correlations (Fig 3G), and reveals that the noise correlations are tuned to a particular time point ($t \approx 0.7$) in a trial (Fig 3F). The main advantage of using copula models [18] is their expressive power in modeling the shape of the relationship between variables, which, in this case, is asymmetric due to the dynamic connections in the neuronal circuit being asymmetric (Fig 3). It was also shown previously that shared excitatory or inhibitory inputs can result in similar non-linear dependencies between neurons [24]. While copula models of the noise correlations do not provide any mechanistic interpretation of these dependencies (e.g. whether neurons are recurrently coupled or share an input), their expressive power allows one to build statistical models for information-theoretic analysis or Bayesian decoding. An additional explicit fully-Bayesian parameterization provided by a GP (Fig 3F) extends the copula model to time-dependent, location-dependent or otherwise continuously parameterized behavioral tasks.

## 3 Validation of mutual information estimation on artificial data

Apart from conditioning on time or on some complex continuous stimuli, Copula-GP models can also provide accurate information estimates. Static copula models are generally well suited for information estimation, as the mutual information between the variables corresponds to the negative entropy of their copula [18, 34]. This also applies to the conditional entropy of our conditional copula models $H(\mathbf{y}|x)$, as it factorizes into marginal and copula components (see Eq (11) in Methods). However, calculating the mutual information between the variable $\mathbf{y}$ and the conditioning variable $x$ is more complicated. We propose two methods for estimating this mutual information $I(\mathbf{y}, x)$: 'integrated' (12) and 'estimated'(13). The first one assumes no conditional dependence in marginals (i.e. $p_i(y_i) = p_i(y_i|x)$ for every $y_i$ and every value of $x$) and only requires a conditional copula model for $p(\mathbf{y}|x)$, which is then integrated using the Monte-Carlo (MC) algorithm (over both $x$ and $\mathbf{y}$) in order to obtain $p(\mathbf{y})$. The second method has no limitations on the conditional dependencies in marginals, but requires an additional copula model for the unconditional distribution $p(\mathbf{y})$ apart for a conditional copula model of $p(\mathbf{y}|x)$ (see Sec. 11 in Methods for details).

We expect our information estimates to be unbiased for every parametric copula model in our framework, due to the MC estimator being unbiased [52]. The only source of bias in our information estimates comes from the accuracy of the model fit obtained via variational inference, which is inevitably biased. This bias, however, is expected to affect 'integrated' and 'estimated' Copula-GP methods differently, since different dependencies are being modeled with the parametric copulas.

In this section, we evaluate the performance of our Copula-GP method and compare it with the other commonly used non-parametric algorithms for mutual information estimation: Kraskov-Stögbauer-Grassberger (KSG [53]), Bias-Improved-KSG by Gao et al. (BI-KSG [54]) and the Mutual Information Neural Estimator (MINE [55]). We construct 3 datasets, which best illustrate the strengths and the limitations of our semi-parametric and other non-parametric methods. These datasets comprise the synthetic samples from relatively low dimensional distributions ($\leq$10D), for which we can still directly calculate the true mutual information. These distributions have uniform marginals, which allows us to compare 'integrated' and 'estimated' approaches.

**Baseline scenario: Gaussian dataset.** First, we consider a dataset sampled from a multivariate Gaussian distribution, with $\mathrm{cov}(y_i, y_j) = \rho + (1 - \rho)\delta_{ij}$, where $\delta_{ij}$ is Kronecker's delta and $\rho = -0.1 + 1.1\,x, x \in [0, 1]$. Our method is expected to perform well on this dataset because the ground truth model is a special case of our parametric copula model. On the other hand, the dataset is expected to expose the limited scalability of non-parametric methods that do not separate the dependence structure from the marginals and do not make any assumptions on the shape of the dependence.

Our Bayesian model selection algorithm (Sec. 6), based on the fully-Bayesian Watanabe–Akaike information criterion (WAIC, see Sec. 6), selected a Gaussian copula on these data, which perfectly matches the true distribution. As a result, we confirm that Copula-GP measures both entropy and mutual information without bias for any number of dimensions in this dataset (within integration tolerance, see Fig 4A; also tested up to $\rho = 0.999$ and up to 20 dimensions in Sec A in S3 Text). The same exact estimation applies to any linear mixture of copulas in our framework as well (Fig 1; see validation on Clayton copula in Sec B in S3 Text), and is covered by the automated tests on simulated data (see S2 Text).

The performance of the non-parametric methods on this dataset is lower: KSG/BI-KSG captured only around 35%-40% of the true mutual information in 10 dimensions, and MINE captured from 70% to 90% depending on the number of hidden units. This agrees with the previous studies, in which KSG and MINE both severely underestimate the MI for high-dimensional Gaussians with high correlation (e.g. see Fig 1 in Belghazi et al. [55]).

**Low entropy scenario: Student T dataset.** Next, we test the Copula-GP performance on the Student T distribution, which can only be approximated by our copula mixtures, but would not exactly match any of the parametric copula families used in our framework (see Fig 1). We keep the correlation coefficient $\rho$ fixed at 0.7, and only change the number of degrees of freedom exponentially: $df = \exp(5x) + 1, x \in [0, 1]$. This makes the dataset particularly challenging for all methods, as all of the mutual information $I(x, \mathbf{y})$ is encoded in tail dependencies of $p(\mathbf{y}|x)$. The true $H(\mathbf{y}|x)$ of the Student T distribution was calculated analytically (see equation A.12 in [56]) and $I(x, \mathbf{y})$ was integrated numerically according to (12) given the true $p(\mathbf{y}|x)$.

Fig 4B shows that most of the methods underestimate $I(x, \mathbf{y})$, while MINE also produces inconsistent results, sensitive to the choice of the number of hidden units. The training curve for MINE with more hidden units (200,500) showed signs of overfitting (abrupt changes in loss at certain permutations) and the resulting estimate was higher than the true $I(x, \mathbf{y})$ at higher dimensions (e.g. 2 times higher for 10 dimensional dataset and 500 hidden units). It was shown before that MINE provides inaccurate and inconsistent results on datasets with low $I(x, \mathbf{y})$ [57]. KSG/BI-KSG methods also failed to estimate information and produced estimates close or even below 0 (mutual information is supposed to be positive). Our 'Integrated' Copula-GP also underestimated the true mutual information by 0.03 bit for every number of dimensions.

For this dataset, we also apply Copula-GP 'estimated', which uses a combination of two copula models for estimating the components of the $I(x, \mathbf{y})$: $H(\mathbf{y})$ and $H(\mathbf{y}|x)$ (see Eq (13)). In lower dimensions, it produces estimates similar to "Copula-GP integrated", but starts overestimating the true MI at higher dimensions, when the inaccuracy of the density estimation for $p(\mathbf{y})$ builds up. This shows the limitation of the "estimated" method, which can either underestimate or overestimate the correct value due to parametric model mismatch, whereas our empirical results show that "integrated" method consistently underestimates the correct value.

From these results, we conclude that, despite the model mismatch, Copula-GP 'integrated' produces imperfect but consistent results in the low mutual information setting, which consistently underestimate the true value by 0.05 bit. At the same time, MINE produced inconsistent

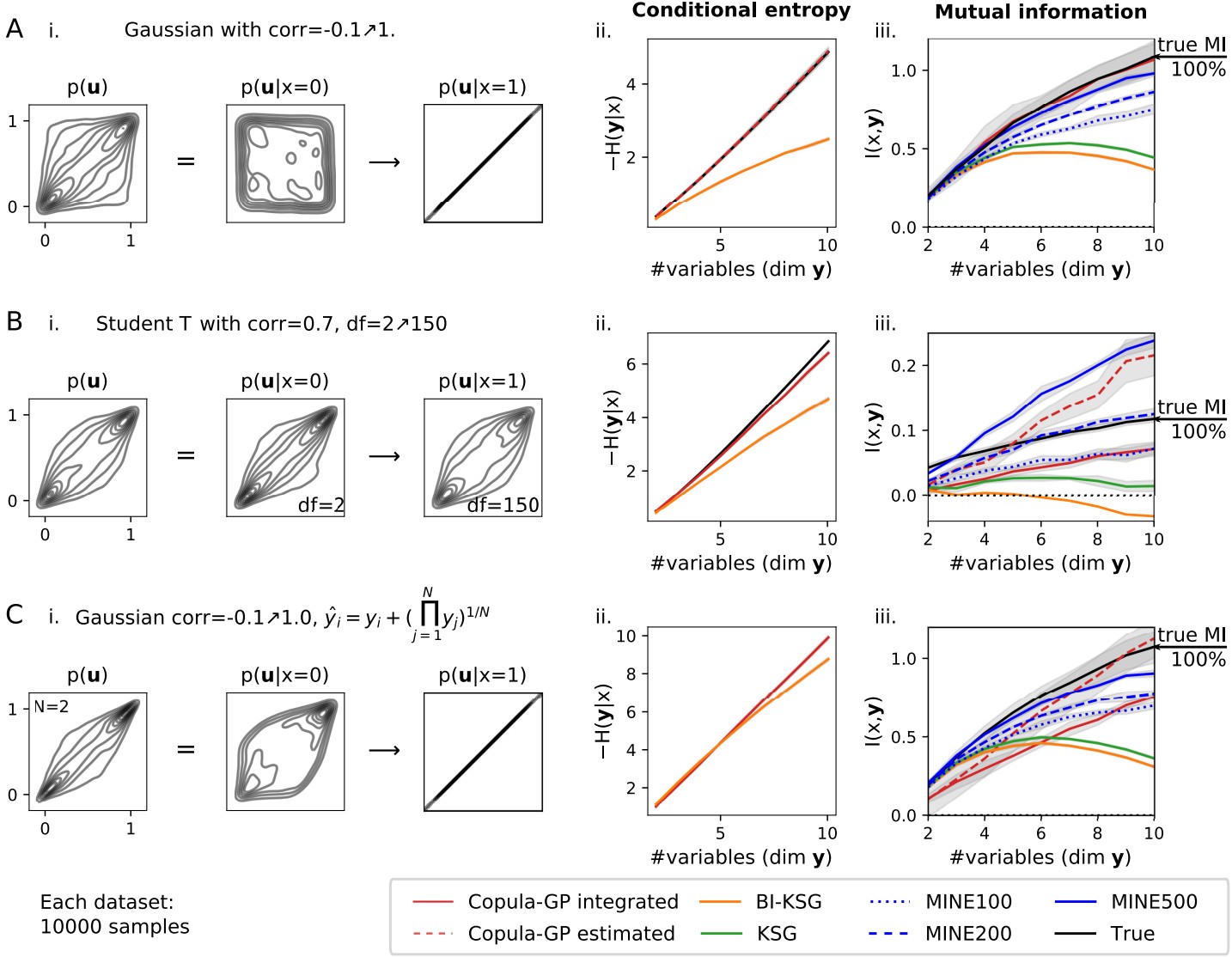

**Fig 4. Comparison of the Copula-GP model against the non-parametric information estimators**, performed on three benchmarking datasets **A** Multivariate Gaussian. **B** Multivariate Student T. **C** Multivariate Gaussian **y** (same as **A**), morphed into another distribution $p(\hat{\mathbf{y}})$ with a tail dependence, while $I(x, \mathbf{y}) = I(x, \hat{\mathbf{y}})$. In each row, the plots show: **i.** the probability density plots from each dataset: the unconditional dependency structure $p(\mathbf{u})$ (left) and conditional dependency structures at the beginning and the end of the parameter domain dom $x = [0, 1]$ (middle and right, respectively). **ii.** conditional entropy $H(\mathbf{y}|x)$; the black line shows the true values, the red line—Copula-GP, the orange line—BI-KSG; Note, that MINE is not included in this comparison, as it does not produce estimates of $H(\mathbf{y}|x)$. **iii.** mutual information $I(x, \mathbf{y})$; black line—true value; red—Copula-GP (solid: MC integration (12); dashed: estimated MI (13)); orange—BI-KSG; green—KSG; blue—MINE (dotted: 100 HU, dashed: 200 HU, solid: 500 HU). Gray intervals show either standard error of mean (SE, 6 repetitions), or $\sqrt{(SE)^2 + (MC_{tol})^2}$ for integrated variables. Note, that MINE estimates are sensitive to the choice of hyper-parameters (e.g. number of hidden units, shown in different line styles).

estimates in high dimensional datasets, ranging from 60% to 200% of the true value, KSG found almost no mutual information between the variables, and BI-KSG even produced invalid negative results.

**No exact model fit scenario: Transformed Gaussian dataset.** Finally, we created a third artificial dataset (Fig 4C) for exploring the limitations of the Copula-GP method. We constructed a distribution, which was drastically different from any of the copula models used in our framework (see Fig 1). We achieved that by applying a homeomorphic transformation **F**

($\mathbf{y}$) to a multivariate Gaussian distribution (from Fig 4A). Since the transformation is independent of the conditioning variable, it does not change the $I(x, \mathbf{y}) = I(x, \mathbf{F}(\mathbf{y}))$ [53]. Therefore, we possess the true mutual information $I(x, \mathbf{y})$, which is the same as for the first example in Fig 4A. Note, however, that there is no ground truth for the conditional entropy in this example, since $H(\mathbf{y}) \neq H(\mathbf{F}(\mathbf{y}))$. We transform the Gaussian copula samples $\mathbf{y} \in [0, 1]^N$ from the first example as $\tilde{y}_i = y_i + (\prod_{j=1}^{N} y_j)^{1/N}$ and again transform the marginals using the empirical probability integral transform $\mathbf{u} = \mathbf{F}(\tilde{\mathbf{y}})$. Both conditional $p(\mathbf{u}|x)$ and unconditional $p(\mathbf{u})$ densities here do not match any of the parametric copulas from Fig 1.

We observe, that the estimates of the non-parametric methods remained almost unchanged after the transformation (compare Fig 4A vs. 4C). Copula-GP, on the other hand, produced erroneous estimates: the 'integrated' method underestimated the true value by 30%, while 'estimated' method overestimated it by just 5%. The MINE estimator (500 hidden units) performed slightly better, capturing 85% of the true MI in 10 dimensions. However, predictions from MINE were sensitive to the choice of hyperparameters (see plots for MINE 100,200 and 500, with estimations ranging from 60% to 200% of the true value in Fig 4B and from 65% to 85% in Fig 4A and 4C).

This result demonstrates that the performance of the parametric Copula-GP model critically depends on the match between the true probability density and the parametric copula model. The mismatch, however, affects the 'integrated' method and the 'estimated' method differently. The 'estimated' method can have either positive or negative bias, depending on which model mismatch had greater impact: either the unconditional ($p(\mathbf{y})$) or the conditional ($p(\mathbf{y}|x)$) copula model in (13), respectively. The 'integrated' method, however, has never overestimated the true value in any of the synthetic datasets. Therefore, while having a higher computational cost, the Copula-GP 'integrated' method can produce more accurate and more reliable results, which either accurately match the true value (Fig 4A) or, when the exact modeling is not possible, consistently underestimate it (Fig 4B and 4C).

**Overall Copula-GP provides accurate and reliable information estimates.**  The examples in Fig 4 show that our method is well suited for estimating the information measures (entropy or mutual information) at higher dimensions. Our Copula-GP provides better information estimates compared to KSG/BI-KSG on all datasets, and also outperforms MINE either when the true distribution matches our parametric model exactly (Fig 4A) or when the absolute values of the mutual information are low (Fig 4B). Even when the exact reconstruction of the density is not possible (e.g. Fig 4B), the mixtures of the copula models are still able to model the changes in tail dependencies, at least qualitatively, and produce reasonable information estimates.

The third dataset in Fig 4C was designed to expose the main limitation of the Copula-GP method: a situation where the parametric copula model is inadequate for the data. Unlike MINE which can produce biased and inconsistent results (e.g. in Fig 4B; also see explanation of this behavior in [57]), our Copula-GP model has never overestimated the mutual information when using the direct 'integrated' Copula-GP method. Most importantly, the performance of the Copula-GP method was less affected by the increase in dimensionality (MINE and KSG estimates diverge from the ground truth at higher dimentsions in Fig 4), suggesting that our semi-parametric method scales better than the non-parametric methods.

## 4 Validation of Copula-GP method on neuronal population activity from the visual cortex of behaving mice

In this section, we investigate the dependencies observed in neuronal and behavioral data and showcase possible applications of the Copula-GP framework. We used two-photon calcium

imaging data of neuronal population activity in the primary visual cortex of mice engaged in a visuospatial navigation task in virtual reality (data from Henschke et al. [16], decontaminated with FISSA algorithm [43]). The visual stimulus in this experiment was uniquely determined by the position in the virtual reality, similarly to the examples in Figs 2 and 3 where the shared input $x$ was determined by time $t$. Briefly, the mice learned to run through a virtual corridor with vertical gratings on the walls (Fig 5A, 0–120 cm) until they reached black walls in the reward zone (Fig 5A, 120–140 cm), where they could get a reward by licking a reward spout as defined in the original publication of this dataset [5].

For demonstration of the suitability of Copula-GP models for neuronal data, we selected one example dataset from an animal in which 102 neurons were imaged in visual cortex, during the first day of the animal's training to the task. We conditioned our Copula-GP model on the position in the virtual environment $x$ and studied the joint distribution of the behavioral ($\tilde{y}_1 \ldots \tilde{y}_5$) and neuronal ($\tilde{y}_6 \ldots \tilde{y}_{109}$) variables (dim $\mathbf{y}$ = 109, Table 1). Fig 5B shows examples of neuronal responses from the selected mouse running along the virtual corridor. The traces show changes in the position $x$ of the mouse as well as the activity of 3 selected neurons and the licking rate.

The goal of our analysis is to build a statistical model of the joint distribution of these variables $p(\mathbf{y}|x)$ and measure how much information these variables carry about each other (i.e. the redundancy) and about the location in the virtual environment. These variables have different patterns of activity depending on $x$ and different signal-to-noise ratios (Fig 5B), which results in drastically different distributions of individual variables (i.e. marginal statistics). For such data, copulas are an uniquely suited tool for 'gluing' these variables together and constructing a statistical model of their joint distribution.

**4.1 Bivariate Copula-GP models find heavy-tailed dependencies in inter-neuronal noise correlations and behavioral modulations.**   We first studied bivariate relationships between the neurons imaged in the visual cortex. In order to do this, we transformed the decontaminated calcium traces (shown in Fig 5B) with a probability integral transform $\mathbf{u}^x = \mathbf{F}(\mathbf{y}|x)$. We observed strong changes in the shape of the dependence $c(\mathbf{u}|x)$ subject to the position in the virtual reality $x$: it was heavy-tailed in some but not all locations (see orange circles in Fig 5C), indicating that the tail-dependence carried information about the location. These heavy-tailed dependencies can be captured by our copula models but would have been missed if the noise correlations were only characterized with a linear (Pearson) correlation coefficient. We applied our Bayesian model selection algorithm (see Sec. 6) to these data, which showed that this dependence structure is best characterized by a combination of Gaussian and Clayton copula (rotated by 90˚). The Clayton copula has a heavy tail, which models the dependence between the extreme values of the variables, e.g. between abnormally high or low single trial responses of these neurons compared to their trial-average responses. This heavy tail can be seen in the probability density plots indicated by orange circles, at the beginning and the end of the virtual corridor (Fig 5C, $x \in [0, 60]$cm and $x \in [140, 160]$cm). The Gaussian copula, on the other hand, has no tail dependence. Therefore, such mixtures of copulas with different tail dependencies allow us to model qualitative changes in the shape of the dependence that we observe in real neuronal data (Fig 5C).

The probability density plots in Fig 5C demonstrate the match between the empirical probability density (outlines) and the copula model density (blue shades) for 4 locations along the virtual corridor. We measure the accuracy of the probability density estimation with the proportion of variance explained $R^2$, which shows how much of the variance of the variable $y_2$ can be predicted given the variable $y_1$ (see (9) in section 10). The average $\overline{R^2}$ for all $y_1$ is provided in the upper right corner of the density plots. Here, for each interval, $\overline{R^2} \geq 94\%$. These results

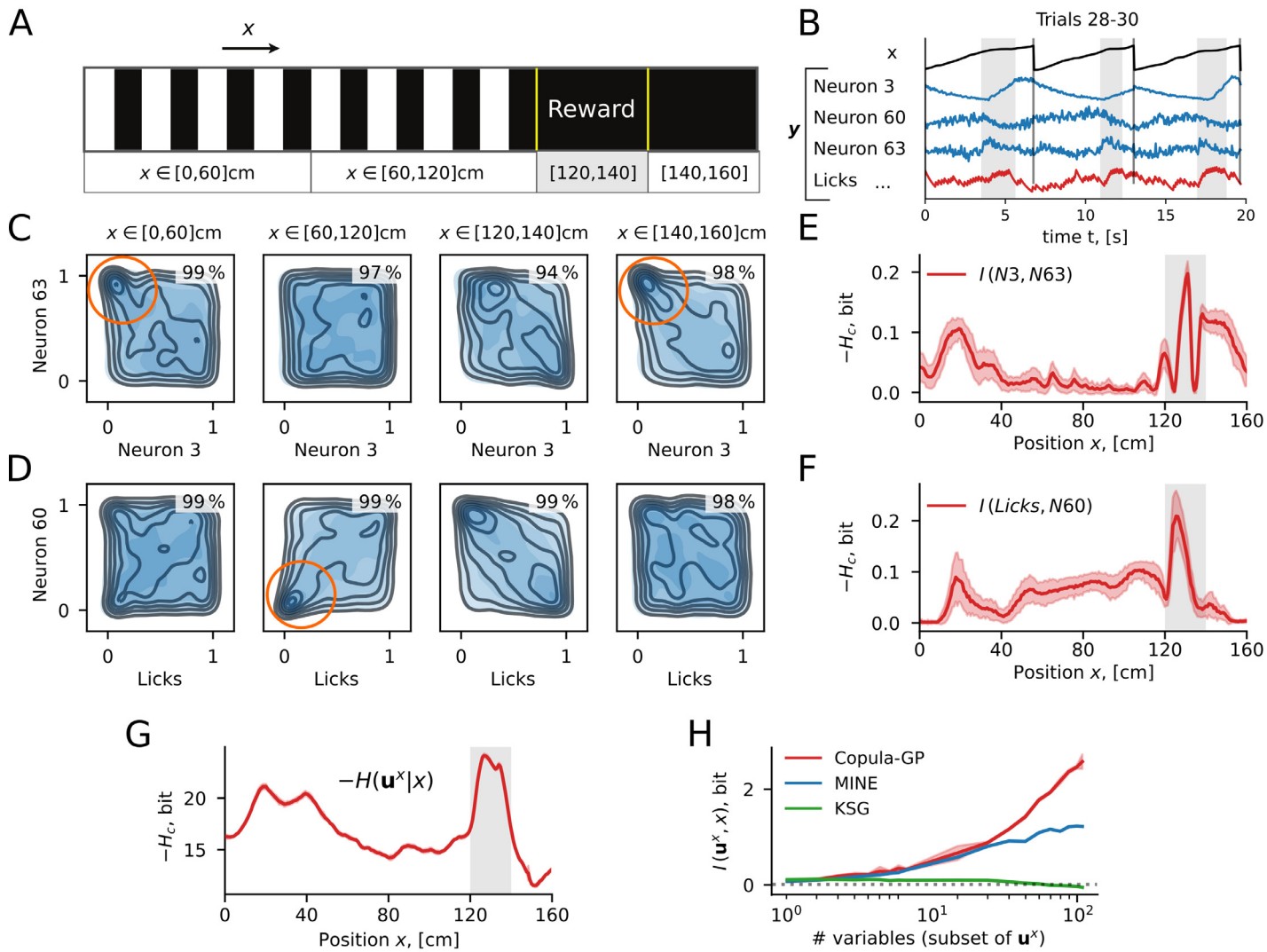

**Fig 5. Validation of Copula-GP method on neuronal population activity and behavioral variables from awake mice.** Copula-GP accurately models the neuronal and behavioral heavy tailed dependencies in the data from the visual cortex of awake mice, and quantifies more mutual information between various combinations of variables than the alternative methods. **A** Schematic of the navigational experimental task [5, 16] in virtual reality; **B** Example traces from ten example trials: $x$ is a position in virtual reality, **y** is a vector of neuronal (blue) and behavioral (red) variables; these traces show that variables have different timescales and different signal-to-noise ratios, which result in different distributions of single variables $y_i$ (i.e. different marginal statistics). **C-D** Copula probability density plots for: the noise correlation between two neurons (number 3 and 63) (C) and for the correlation between one neuronal activity (60) vs. one behavioral variable (licks) (D); Black outlines show empirical copula, shades of blue—the best fitting Copula-GP model: a mixture of Gaussian + 90˚-rotated Clayton copula in (C) and a mixture of Frank + 0˚-rotated Clayton + 270˚-rotated Gumbel copula in (D) (see S4 Text for model parameters). Similarly to the example with two dynamically coupled neurons (Fig 3G), these copulas are heavy-tailed. The goodness-of-fit for these models is measured with the proportion of the variance explained $\overline{R^2}$, which is indicated in the upper-right corner of each plot corresponding to a range of positions in virtual reality; **E-G** Conditional entropy for the bivariate examples (E-F) and the population-wide statistics (G) all peak in the reward zone; this entropy is equivalent to the mutual information between variables, given the position $x$, which means that the variables carry the most information about each other when the animal is in the reward zone. **H** Comparison of Copula-GP method ("integrated") vs. non-parametric MI estimators (MINE [55] and KSG [53]) on estimating the amount of information about the location $x$ from the subsets of variables $\mathbf{u}^x$. While the true $I(x, \mathbf{u}^x)$ is unknown, the validation on synthetic data (Fig 4) suggest that Copula-GP "integrated" does not overestimate the amount of mutual information. Yet, Copula-GP "integrated" quantifies more information about the position $x$ from the large subsets of data $\mathbf{u}^x$ than MINE and KSG methods.

suggest that such heavy-tailed noise correlations between neurons can be captured by our Copula-GP model.

Next, we show that our model can be applied not only to the neuronal data, but also to any of the behavioral variables. Fig 5D shows the dependence structure between one of the neurons

**Table 1. A list of variables in the navigational task dataset from Pakan et al.; for a detailed description of the task and the reward, see the original paper [5].** The variables are grouped according to their type; their order $\tilde{y}_1 \ldots \tilde{y}_{109}$ does not correspond to the order of variables $y_1 \ldots y_{109}$ in vine models (see text).

| Variable | Name | Type | Units |
|:---:|:---:|:---:|:---:|
| $x$ | Position | Determines visual stimulus | cm |
| $\tilde{y}_1$ | Velocity | Behavioral | cm/s |
| $\tilde{y}_2$ | Licks | | 1/s |
| $\tilde{y}_3$ | Early reward | | |
| $\tilde{y}_4$ | Default reward | | |
| $\tilde{y}_5$ | Any reward | | |
| $\tilde{y}_6$ | Global neuropil | Neuronal | $[1] = \Delta F/F$ |
| $\tilde{y}_7$ | Background fluorescence | | |
| $\tilde{y}_8$ | Neuron 1 | | |
| $\ldots$ | $\ldots$ | | |
| $\tilde{y}_{109}$ | Neuron 102 | | |

and the licking rate. The best selected mixture model here is Frank + Clayton 0˚ + Gumbel 270˚, which again provides an accurate estimate of the conditional dependence between the variables ($\overline{R^2} \geq 98\%$). This statistical model describes how the activity of neuron 60 is modulated be the licking behavior of the animal, showing how both variables jointly deviate from their trial-averaged values. For example, these variables were negatively correlated in the reward zone, while before the reward zone, they were positively correlated with a lower tail dependence (orange circle in Fig 5). Thus, our model can describe not only the noise correlations, but also the behavioral modulations of neuronal activity.

These examples illustrate that copula mixtures in the Copula-GP model provide an accurate fit for the pairwise joint distributions and make the shape of the dependence explicit with a direct visualization (Fig 5C and 5D). Both the noise correlations between two neurons and the dependencies between neurons and behavioral variables have heavy tails (see orange circles in Fig 5C and 5D). The heavy tails in neuronal noise correlations were expected, based on previous studies that considered various excitatory or inhibitory shared inputs into pairs of neurons [24]. Although, we also found unexpected qualitative changes of the dependence from one location to another, which could even change from a positive to a negative correlation (e.g. Fig 5D [60, 120] vs. [120, 140] cm). The parameters of the linear mixtures can qualitatively describe these changes in shape of the dependence, since the mixing coefficients identify the contribution of different copula components at any given location (see S4 Text for details and examples). Most importantly, the accurate approximation of the copula density, provided by mixture models, allows us to use them for estimating the mutual information between neuronal and behavioral variables.

**4.2 Copula-GP reveals behaviorally-relevant locations without prior knowledge of task structure.** Fig 5E and 5F show the negative conditional entropy $-H(\mathbf{u}^x|x)$, which is equivalent to the mutual information between two variables $I(y_i, y_j|x)$ given the position $x$. The confidence intervals (shaded area) were obtained using samples from the Gaussian Process posterior, which reflects the uncertainty in model parameters. For both examples, the MI between variables (between two neurons in Fig 5E and a neuron vs. the licking rate in Fig 5F) peaks in the reward zone located at 120–140 cm in this task (Fig 5A). The bivariate Copula-GP models were agnostic of the reward mechanism in this task, yet they reveal a location that is strongly

encoded in the neuronal population, and appears in our analysis as an anomaly in the mutual information between neuronal and behavioral variables.

This analysis shows that the Copula-GP model can reveal the behaviorally-relevant locations in the virtual corridor without any prior knowledge about the task structure by quantifying the information encoded in non-linear dependencies between single trial neuronal responses.

**4.3 Copula-GP captures more information from a large neuronal population compared to non-parametric estimators.**   Finally, we constructed a C-vine model describing the distribution between all neuronal and behavioral variables in the dataset ($\{u_1^x...u_{109}^x\}$, dim $\mathbf{u}^x = 109$, see Table 1). The vine model decomposes the high-dimensional distribution into a hierarchy of bivariate copulas, where each level of the hierarchy (a vine tree) adds a conditional dependence on one of the variables. Such factorization allows one to escape the 'curse of dimensionality' while still accounting for the higher-order correlations between the variables [23].

Using this C-vine model, we have calculated the negative conditional entropy, which is equivalent to the *generalized redundancy* between all variables given the position $x$: $-H(\mathbf{u}^x|x) = \sum H(\tilde{y}_i|x) - H(\tilde{y}_1, \ldots, \tilde{y}_{109}|x)$. The generalized redundancy in Fig 5G peaks in the reward zone (similarly to Fig 5E and 5F) and also at the beginning of the trial, where the velocity of the animal varied the most on a trial-to-trial basis. In those regions where the redundancy was high, a larger portion of neural variability can be accounted for, based on the activity of the other neurons and on the current behavior of the animal.

We next studied the complexity of the trained vine copula model. In this conditional C-vine trained on a neuronal dataset with 109 variables (Table 1), 5253 out of 5886 (= $m(m-1)/2$, 89%) bivariate models were Independence copula, which leaves only 633 non-Independence copula models. These remaining 633 non-independence copulas still describe a distribution with a relatively high intrinsic dimensionality. We can estimate this dimensionality by, for example, finding the minimal number of copula vine trees to capture 90% of the entropy $H(\mathbf{y}|x)$. For this complex dataset with a self-paced navigational task, this estimate equals 17 dimensions, which is much lower than the dimensionality of the recorded variables (102 neurons, 109 neuronal and behavioral variables). Such sparsity of the C-vine arises from the low dimensionality of the experimental tasks. Therefore, for our data-driven model and for typical neuronal population recordings, the time required for mutual information estimation depends mostly on the dimensionality of the task rather than on the number of recorded variables (see Sec. 8 in Methods).

Next, we compare our Copula-GP against the non-parametric methods in estimating the mutual information $I(x, \mathbf{u}^x)$ between the position of the animal in virtual reality $x$ and the trial-to-trial variability of the recorded neuronal and behavioral variables $\mathbf{u}^x$. While constructing the C-vine, we ordered the neuronal variables according to their pairwise rank correlations (see Sec. 7). We then considered subsets of the first $N$ variables and measured the mutual information (MI) between each subset of neuronal responses and the position in virtual reality. We compared the performance of our Copula-GP method on these subsets of $\mathbf{u}^x$ vs. KSG and MINE. Fig 5H shows that all 3 methods provide similar results on subsets of up to 10 variables, yet in higher dimensions both MINE and KSG show smaller amount of information $I(x, \{u_{i<N}^x\})$ compared to our Copula-GP method (47% and 0% of the Copula-GP prediction, respectively), which qualitatively agrees with the results obtained on the synthetic data (Fig 4). The true values of $I(x, \{u_{i<N}^x\})$ are unknown, yet we expect the integrated Copula-GP to underestimate the true value (solid red line in Fig 4 above). Even this estimate, however, exceeds the MINE estimate by 211%, suggesting that both non-parametric methods (KSG and MINE) severely underestimate the mutual information $I(x, \{u_{i<N}^x\}$ in a large neuronal

population. These results demonstrate superior performance of our Copula-GP model on information estimation in high-dimensional neuronal data.

## Comparison with alternative methods

Throughout the paper, we compare our model with the state-of-the-art general-purpose information estimators, which were also recently applied to the neuronal data [58, 59]. We show, that both of the state-of-the-art information estimators—KSG and MINE—have poor performance on high dimensional neuronal data. Intuitively, the difference in performance between our parametric method and these non-parametric methods can be explained as follows. The estimators based on k-nearest neighbor distances, such as KSG [53] / BI-KSG [54] rely on the volumes of the nearest neighborhoods of the samples for density estimation, which expand with the increase in the number of dimensions, resulting in poor information estimates. The neural network approaches, such as MINE [55], provide better scalability, but have many parameters and require high sample numbers in order to achieve a given accuracy with a given confidence. As a result, both classes have poor performance when the number of samples (trials) is low and the dimensionality (number of neurons and behavioral variables) is high. To overcome this 'curse of dimensionality', some assumptions about the dependency between variables are required [60].

Our Copula-GP method follows this approach in that it assumes certain shapes of the dependence between variables by specifying parametric copula distributions (see Fig 1). These constraints on the shape of the dependence, however, are less restrictive than the assumptions in other commonly used models, such as maximum entropy models [61]. Some of these maximum entropy models utilized conditioning on the overall population activity in order to take some of the higher-order correlations into account [62–64]. These models resemble the conditioning used in the first tree of our vine copulas, especially when we select the global neuropil activity as the first conditioning variable (see Sec. 7). However, all these population tracking maximum entropy models assumed linear coupling of each individual neuron with the population rate, characterized by a single parameter (thus, $\sim N^2$ parameters in total). We relax this constraint by introducing mixtures of copulas from different copula families, which model complex nonlinear pairwise dependencies between variables. The complexity of the copula mixture 'adapts' to the complexity of the data, since we select the least complex model that can capture the dependence via the information criterion (WAIC). As a result, our semi-parametric Copula-GP model takes the best of both worlds, performing well given a limited number of high-dimensional samples, and, at the same time, being more flexible in representing the conditional dependencies than maximum entropy models.

Alternatively to the information-theoretic approaches, some popular non-copula methods could model the neuronal responses in the aforementioned navigational task dataset. These methods include GPFA or GLMs on the deconvolved spikes [65]. Also, for the multi-photon calcium recordings (ΔF/F), the newly developed ValPACa method could, perhaps, be an alternative [66]. These models, however, require certain assumptions on the marginal statistics (e.g. Poisson or Gaussian), and only model the joint distribution of the neuronal activity (without behavioral variables). Contrary to these approaches, copula methods allow us to model the dependencies between neuronal and behavioral variables, combining the elements with utterly different statistics (e.g. licks or velocity with neuronal activity). In addition, Copula-GP explicitly represents the dependencies as a function of time or position. As a result, the conditional entropy calculated with our model (Fig 5E–5G) revealed insightful information about the task structure, highlighting the location of the reward zone. To the best of our knowledge, there are currently no other methods with this combination of properties that include: explicit

conditioning on stimuli, flexibility in probability density estimation, scalability and accuracy in information estimation.

## Discussion

We have developed the Copula-GP framework for modeling neuronal and behavioral dependencies. The validation of our method on synthetic and neuronal imaging data demonstrates that Copula-GP can provide accurate statistical models and quantifies most of the mutual information between neuronal and behavioral variables from experimental recordings. Unlike other statistical models commonly used in neuroscience, Copula-GP can merge behavioral variables along with neuronal responses into a joint statistical model. The parameterization with a Gaussian process makes our framework particularly useful for modeling neuronal dynamics (as in the example in Fig 3) or for visualizing the conditional information in neuronal populations (e.g. the representation of the reward zone in Fig 5E–5G). Apart from time and space, the model can be conditioned on any other continuous variable, such a phase, velocity, direction, orientation, etc. If this variable also uniquely determines the stimulus, then the components of the model (2) can be interpreted in terms of 'noise correlations', approximated by a parametric copula.

The main advantage of using copula models is their ability to 'glue' together the variables with drastically different statistics. This allowed us to use the calcium traces ($\Delta F/F$), instead of the deconvolved spikes, together with a variety of behavioral variables that are measured in different units (Table 1) and have different marginal distributions. Also, since our framework does not rely on spike counts, no binning of the data along the spatial locations of the virtual environment was required. Instead, the Gaussian process component of our Copula-GP model could account for the unevenly distributed data-points. This variability in the amount of available data at different locations contributes to the uncertainty estimates in Fig 5E and 5F (confidence intervals for the MI estimates are shown in red shades and have variable size). As a result, Copula-GP can be applied directly to calcium imaging data from the complex (e.g. self-paced) tasks, without the need for spike inference and binning.

The combination of parametric copula models with GP priors has been previously applied to weather forecasts, geological data, or stock market data [30, 31]. Yet, this Copula-GP approach has never been applied to neuronal recordings before. As a distinct feature in neuronal datasets that limited the application of the original method, we observed changes in tail dependencies with regard to the conditioning variable. None of the single parametric copula models could represent such dependencies (Fig 1). Therefore, the main contributions of our method are the following: 1) we applied gaussian process priors on both the copula parameters and the mixing concentrations in the copula mixture models (see Sec. 5); 2) we have developed the algorithms for constructing such copula mixtures with a Bayesian model selection algorithm (see Sec A in S1 Text and S2 Text); 3) we have proposed two approaches for mutual information estimation and validated them against the state-of-the-art information estimators. These contributions made the method flexible enough to describe the dependencies between neurons (Fig 5C and 5D and A in S4 Text) and applicable for information-theoretic analysis of neural codes (Fig 5E–5H).

Apart from these methodological contributions that made the Copula-GP method applicable to neuronal data, we have also ensured that the method is scalable to large neuronal datasets. We overcame the well-known poor scalability of Gaussian processes by using approximate methods and by parallelizing the parameter inference of different pairwise copulas in a vine tree on a GPU, building our package based upon the PyTorch [67] and GPyTorch [68] libraries. We demonstrated that the model scales well at least up to 109 variables and 21k

samples, while theoretically, the parameter inference scales as $\mathcal{O}(n \cdot m^2)$, where $n$ is the number of samples and $m$ is the (effective) number of variables (see Sec. 8). Such scaling with the effective number of dimensions in the data results from the sparsity of the vine copula models. This sparsity also allows to use the trained models for decoding without access to high-performance computing infrastructure. The implementation of our method is available on Github: https://github.com/NinelK/CopulaGP.

We have validated the Copula-GP method on mutual information estimation for synthetic datasets and for neuronal and behavioral recordings. Our method, therefore, can be readily used for information-theoretic analysis. We have shown how our method discovers the structure of the task in an unsupervised way based on one day of recordings. The same analysis, applied to multiple days, can detect how the amount of information in the neuronal population about a certain behaviorally-relevant variable changes during training of the animal. The estimates from different days can be compared, and the uncertainty of the estimated information allows for statistical hypothesis testing. Such analysis does not require chronic recordings of the same neurons across days, as it focuses on the information instead of particular neuronal codes. A similar analysis can be applied to finding the mutual information between the responses in different brain areas. As a fully-Bayesian model, Copula-GP can also guide data collection [69], suggesting whether the number of recorded neurons or behavioral variables is sufficient for confirmation or rejection of a hypothesis. A visualization similar to Fig 5E–5H can also highlight the areas, where additional data collection is needed.

Apart from a mutual information estimator, our Copula-GP model can be viewed as a general probabilistic model of the distribution of **y** given $x$. This distribution can be used for Bayesian decoding. The Bayesian decoding is complimentary to the mutual information between the stimulus and the response: while the former predicts the most likely stimulus that caused the observed neuronal responses, the latter quantifies the overall knowledge about the stimuli contained in single-trial responses [20]. The application of such decoders is not limited to basic science, but can be extended to brain-computer interfaces, where Gaussian Process based methods were already successfully applied [38]. Our Copula-GP model can potentially be adapted as an observation model for Gaussian Process Latent Variable Models [70], as a more flexible statistical model of neuronal responses.

Future work will focus on implementing model selection for the vine structure, incorporating discrete variables (as in Onken et al. [23]) and improving the scalability of the mutual information estimation algorithm. Another interesting possible extension involves a recently developed multi-output non-reversible kernel [71], which can make the Copula-GP model more suitable for modeling non-reversible dynamics in neuronal populations. These modifications would further broaden the applicability of the Copula-GP method, which, in its current state, is best suited for analyzing the activity in large neuronal populations measured with calcium imaging methods.

In summary, we demonstrated that the Copula-GP approach can make stochastic relationships (e.g. noise correlations) explicit and accurately model dependencies between neuronal responses, sensory stimuli, and behavioral variables.

# Methods

## 5 Parametric copula mixtures with Gaussian process priors

In this paper, we use semi-parametric approach: non-parametric marginals and parametric copulas. We used the two-stage inference for margins (IFM) training scheme, which is typically used high dimensional datasets [72]. First, univariate marginals were estimated and used

**Table 2. Bivariate copula families and their GPLink functions.**

| Copula | Domain | GPLink($f$) : $\mathbb{R} \rightarrow \mathbf{dom}(c_j)$ |
|---|---|---|
| Independence | – | – |
| Gaussian | $[-1,1]$ | Erf($f$/1.4) |
| Frank | $(-\infty, \infty)$ | $0.1 \cdot f + \text{sign}(f) \cdot (0.1 \cdot f)^2$ |
| Clayton | $[0,\infty)$ | $\exp(0.2 \cdot f)$ |
| Gumbel | $[1,\infty)$ | $1 + \exp(0.1 \cdot f)$ |

to map the data onto a multidimensional unit cube. Second, the parameters of the copula model were inferred.

**Bivariate copula families.** We use 4 copula families as the building blocks for our copula models: Gaussian, Frank, Clayton and Gumbel copulas (Fig 1). All of these families have a single parameter, corresponding to the rank correlation (Table 2). We also use rotated variants (90˚, 180˚, 270˚) of Clayton and Gumbel copulas in order to express upper tail dependencies and negative correlation.

Since we are primarily focused on the analysis of neuronal data, we have first visualized the dependencies in calcium signal recordings after a probability integral transform, yielding empirical conditional copulas. As a distinct feature in neuronal datasets, we observed changes in tail dependencies with regard to the conditioning variable. Since none of the aforementioned families alone could describe such conditional dependency, we combined multiple copulas into a linear *mixture model*:

$$c(\mathbf{u}|x) = \sum_{j=1}^{M} \phi_j(x) c_j(\mathbf{u}; \theta_j(x)), \tag{3}$$

where $M$ is the number of elements, $\phi_j(x)$ is the concentration of the $j$th copula in a mixture, $c_j$ is the pdf of the $j$th copula, and $\theta_j$ is its parameter.

Each of the copula families includes the Independence copula as a special case. To resolve this overcompleteness, we add the Independence copula as a separate model with zero parameters (Table 2). For independent variables $\mathbf{y}_{ind}$, the Independence model will be preferred over the other models in our model selection algorithm (Sec. 6), since it has the smallest number of parameters.

**Gaussian process priors.** We parametrize each copula in the mixture model with an independent latent GP: $\mathbf{f} \sim \mathcal{N}(\mu \times \mathbf{1}, K_\lambda(x, x))$. For each copula family, we constructed GPLink functions (Table 2) that map the GP variable onto the copula parameter domain: $\theta_j = \text{GPlink}_{c_j}(f_j), \mathbb{R} \rightarrow \text{dom}(c_j)$. Next, we also use GP to parametrize concentrations $\phi_j(x)$, which are defined on a simplex ($\Sigma \phi = 1$):

$$\phi_j = (1 - t_j) \prod_{m=1}^{j-1} t_m, \quad t_m = \Phi\left(\tilde{f}_m + \Phi^{-1}\left(\frac{M - m - 1}{M - m}\right)\right), \quad t_M = 0,$$

where $\Phi$ is a CDF of a standard normal distribution and $\tilde{\mathbf{f}}_\mathbf{m} \sim \mathcal{N}(\tilde{\mu}_m \times \mathbf{1}, \tilde{K}_{\tilde{\lambda}_m}(x, x))$. We use the RBF kernel $K_\lambda(x, x)$ with bandwidth parameter $\lambda$. Therefore, the whole mixture model with $M$ copula elements is parameterized by $[2\,M - 1]$ independent GPs and requires $[2\,M - 1]$ hyperparameters: $\{\lambda\}_M$ for $\boldsymbol{\theta}$ and $\{\tilde{\lambda}\}_{M-1}$ for $\phi$.

**Approximate inference.** Since our model has latent variables with GP priors and intractable posterior distribution, direct maximum likelihood Type-II estimation is not possible and

an approximate inference is needed. Such an inference problem with copula models has previously been solved with the expectation propagation algorithm (Hernández-Lobato et al. [31], see a direct comparison with our method in Sec C in S3 Text), which was not suitable for large scale data. Recently, a number of scalable approximate inference methods were developed, including stochastic variational inference (SVI) [73, 74], scalable expectation propagation (SEP) [75], and MCMC based algorithms [76], as well as a scalable exact GP [77]. We chose to use SVI due to availability of the well-established GPU-accelerated libraries: PyTorch [67] and GPyTorch [68].

In particular, we used stochastic variational inference (SVI) with a single evidence lower bound [78]:

$$\mathcal{L}_{\text{ELBO}} = \sum_{i=1}^{N} \mathbb{E}_{q(f_i)} \left[ \log p(y_i | f_i) \right] - \text{KL}[q(\mathbf{u}) || p(\mathbf{u})], \tag{4}$$

implemented as `VariationalELBO` in GPyTorch [68]. Here $N$ is the number of data samples, $\mathbf{u}$ are the inducing points, $q(\mathbf{u})$ is the variational distribution and $q(\mathbf{f}) = \int p(\mathbf{f}|\mathbf{u})q(\mathbf{u})d\,\mathbf{u}$.

Following the Wilson et al. [79] approach (KISS-GP), we then constrain the inducing points to a regular grid, which applies a deterministic relationship between $\mathbf{f}$ and $\mathbf{u}$. As a result, we only need to infer the variational distribution $q(\mathbf{u})$, but not the positions of $\mathbf{u}$. The number of grid points is one of the model hyper-parameters: `grid_size`. Note that while approximate inference with KISS-GP uses a discrete set of inducing points (pseudo-inputs), it does not imply discretization of the input variable. The Gaussian process still uses a continuous kernel and assumes a continuous input.

Eq (4) enables joint optimization of the GP hyperparameters (constant mean $\mu$ and two kernel parameters: scale and bandwidth) and parameters of the variational distribution $\mathbf{q}$ (mean and covariance at the inducing points: $\mathbf{u} \sim \mathcal{N}(\mu_u \times \mathbf{1}, \Sigma_u)$) [78]. We have empirically discovered by studying the convergence on synthetic data, that the best results are achieved when the learning rate for the GP hyperparameters (`base_lr`) is much greater than the learning rate for the variational distribution parameters (`var_lr`, see Table 3).

**Priors**. For both the neuronal and the synthetic data, we use a standard normal prior $p(\mathbf{u}) \sim \mathcal{N}(\mathbf{0}, I)$ for a variational distribution. Note, that the parametrization for mixture models was chosen such that the aforementioned choice of the variational distribution prior with zero mean corresponds to *a priori* equal mixing coefficients $\phi_j = 1/M$ for $j = 1\ldots M$. In our experiments with the simulated and real neuronal data, we observed that the GP hyper-parameter optimisation problem often had 2 minima (which is a common situation, see Fig 5.5 on page 116 in [80]). One of those corresponds to a short kernel lengthscale ($\lambda$) and low noise ($\min_{\mathbf{f}} \sigma^2$), which we interpret as overfitting. To prevent overfitting, we used $\lambda \sim \mathcal{N}(0.5, 1.0)$ prior on RBF kernel lengthscale parameter.

**Table 3. Hyper-parameters of the bivariate Copula-GP model.**

| Hyper-parameter | Value | Description |
|---|---|---|
| `base_lr` | 0.05 | Learning rate for GP parameters |
| `var_lr` | 0.02 | Learning rate for variational distribution |
| `grid_size` | 60 | Number of inducing points for KISS-GP |
| `waic_tol` | 0.005 | Tolerance for WAIC estimation |
| `loss_tol` | $10^{-4}$ | Loss tolerance that indicates the convergence |
| `check_waic` | 0.005 | Loss tolerance when we check WAIC |

... and GPLink parameters listed in Table 1.

**Optimization**. We use the Adam optimizer with two learning rates for GP hyper-parameters (`base_lr`) and variational distribution parameters (`var_lr`). We monitor the loss (averaged over 50 steps) and its changes in the last 50 steps: `∆ loss = mean(loss[-100:-50])−mean(loss[-50:])`. If the change becomes smaller than `check_waic`, then we evaluate the model WAIC and check if it is lower than $-\text{WAIC}_{tol}$. If it is higher, we consider that either the variables are independent, or the model does not match the data. Either way, this indicates that further optimisation is counterproductive. If the WAIC $< -\text{WAIC}_{tol}$, we proceed with the optimisation until the change of loss in 50 steps $\Delta loss$ becomes smaller than `loss_tol` (see Table 3).

**Hyper-parameter selection**. The hyper-parameters of our model (Table 3) were manually tuned, often considering the trade off between model accuracy and evaluation time. A more systematic hyper-parameter search might yield improved results and better determine the limits of model accuracy.

## 6 Bayesian model selection

We use the Watanabe–Akaike information criterion (WAIC [81]) for model selection. WAIC is a fully Bayesian approach to estimating the Akaike information criterion (AIC) (see (31) in the original paper by Watanabe (2013) [81]). The main advantage of the method is that it avoids the empirical estimation of the effective number of parameters, which is often used for approximation of the out-of-sample bias. It starts with the estimation of the log pointwise posterior predictive density (lppd) [82]:

$$\widehat{\text{lppd}} = \sum_{i=1}^{N} \log\left(\frac{1}{S}\sum_{s=1}^{S} p(y_i|\theta^s)\right), \qquad p_{\text{WAIC}} = \sum_{i=1}^{N} V_{s=1}^{S}(\log p(y_i|\theta^s)),$$

where $\{\theta^s\}_S$ is a draw from a posterior distribution, which must be large enough to represent the posterior. Next, the $p_{\text{WAIC}}$ approximates the bias correction, where $V_{s=1}^{S}$ represents sample variance. Therefore, the bias-corrected estimate of the log pointwise posterior predictive density is given by:

$$e\widehat{\text{lppd}}_{\text{WAIC}} = \text{lppd} - p_{\text{WAIC}} = -N \cdot \text{WAIC}_{original}.$$

In the model selection process, we aim to choose the model with the lowest WAIC. Since our copula probability densities are continuous, their values can exceed 1 and the resulting WAIC is typically negative. Zero WAIC corresponds to the Independence model (pdf = 1 on the whole unit square).

Since the total number of combinations of 10 copula elements (Fig 1, considering rotations) is large, exhaustive search for the optimal model is not feasible. In our framework, we propose two model algorithms for constructing close-to-optimal copula mixtures: *greedy* and *heuristic* (see S1 Text for details). The greedy algorithm is universal and can be used with any other copula families without adjustment, while the heuristic algorithm is fine-tuned to the specific copula families used in this paper (Fig 1). Both model selection algorithms were able to select the correct 1- and 2-component model on simulated data and at least find a close approximation (within $\text{WAIC}_{tol} = 0.005$) for more complex models (see validation of model selection in S2 Text).

## 7 Copula vine constructions

High-dimensional copulas can be constructed from bivariate copulas by organizing them into hierarchical structures called *copula vines* [21]. There are many possible decompositions based

on different assumptions about conditional independence of specific elements in a model, which can be classified using graphical models called *regular vines* [83, 84]. A regular vine can be represented using a hierarchical set of trees, where each node corresponds to a conditional distribution function (e.g. $F(u_2|u_1)$) and each edge corresponds to a bivariate copula (e.g. $c(u_2, u_3|u_1)$). The copula models from the lower trees are used to obtain new conditional distributions (new nodes) with additional conditional dependencies for the higher trees, e.g. a `ccdf` of a copula $c(u_2, u_3|u_1)$ and a marginal conditional distribution $F(u_2|u_1)$ from the 1st tree provide a new conditional distribution $F(u_3|u_1, u_2)$ for a 2nd tree. Therefore, bivariate copula parameters are estimated sequentially, starting from the lowest tree and moving up the hierarchy. The total number of edges in all trees (= the number of bivariate copula models) for an $m$-dimensional regular vine equals $m(m - 1)/2$.

The regular vines often assume that the conditional copulas $c(u_i, u_j|\{u_k\})$ themselves are independent of their conditioning variables $\{u_k\}$, but depend on the them indirectly through the conditional distribution functions (nodes) [85]. This is known as the *simplifying assumption* for vine copulas [86], which, if applicable, allows to escape the curse of dimensionality in high-dimensional copula construction.

In this study, we focus on the *canonical vine* or *C-vine*, which has a unique node in each tree, connected to all of the edges in that tree. It factorizes the high-dimensional copula probability density function as follows:

$$c(\mathbf{u}) = \left[\prod_{i=2}^{N} c_{1i}(u_1, u_i)\right] \times \left[\prod_{i=2}^{N}\prod_{j=i+1}^{N} c_{ij|\{k\}_{k<i}}(F(u_i|\{u_k\}_{k<i}), F(u_j|\{u_k\}_{k<i}))\right] \qquad (5)$$

where $\{k\}_{k<i} = 1, \ldots, i - 1$ and $F(.|.)$ is a conditional CDF. For graphical illustration, see, for example, Fig 2 in Aas et al. [21]. Note, that all of the copulas in (5) can also be conditioned on $x$ via Copula-GP model.

The C-vine was shown to be a good choice for neuronal datasets [23], as they often include some proxy of neuronal population activity as an outstanding variable, strongly correlated with the rest. This variable provides a natural choice for the first conditioning variable in the lowest tree. In the neuronal datasets from [16], this outstanding variable is the global fluorescence signal in the imaged field of view (global neuropil, variable $\tilde{y}_6$ in Table 1).

To construct a C-vine for describing the neuronal and behavioral data from [16], we used a heuristic element ordering based on the sum of absolute values of Kendall's $\tau$ of a given element with all of the other elements. It was shown by Czado et al. [36] that this ordering facilitates C-vine modeling. For all of the animals and most of the recordings (14 out of 16) the first variable after such ordering was the global neuropil activity. This again confirms, that a C-vine with the global neuropil activity as a first variable is an appropriate model for the dependencies in neuronal datasets.

**Number of parameters in a C-vine model.** A full C-vine from $m$ variables comprises $m \cdot (m - 1)/2$ bivariate copulas. We model bivariate copulas with copula mixtures. As we mentioned above, each copula mixture has $\max(0, 2M - 1)$ parameters, where $M$ is the number of mixture components ($M = 0$ here corresponds to Independence model, which does not have parameters; the maximal number of mixing components is $M_{max} = 5$). Therefore, depending on the data, the actual number of parameters can vary greatly (from 0 to $(2M_{max} - 1) \cdot m \cdot (m - 1)/2$).

## 8 Algorithmic complexity

In this section, we discuss the algorithmic complexity of the parameter inference for a C-vine copula model.

The parameter inference for each of the bivariate Copula-GP models scales as $\mathcal{O}(n)$, where $n$ is the number of samples, since we use a scalable kernel interpolation KISS-GP [79]. As we mentioned in Sec. 7, a full $m$-dimensional C-vine model requires $m(m-1)/2$ bivariate copulas, trained sequentially. As a result, the $\mathcal{O}(n)$ GP parameter inference has to be repeated $m(m-1)/2$ times, which yields $\mathcal{O}(n \cdot m^2)$ complexity.

In practice, the computational cost (in terms of time) of the parameter inference for each bivariate model varies from tens of seconds to tens of minutes. The heuristic model selection is designed in such a way, that it discards independent variables in just around 20 seconds (line 3 in Alg. 2). As a result, most of the models are quickly skipped and further considered as Independence models. When the model is evaluated, the Independence components are also efficiently 'skipped' during sampling, as `ppcf` function is not called for them. The Independence models also add zero to C-vine log probability, so they are also 'skipped' during log probability calculation. They also reduce the total memory storage, as no GP parameters, which predominate the memory requirements, are stored for these models.

In practice, this means that the algorithmic complexity of the model is much better than the naïve theoretical prediction $\mathcal{O}(n \cdot m^2)$, based on the structure of the graphical model. Suppose that the actual number of the non-Independence models $N_{nI}$ in a vine model is much smaller than $m(m-1)/2$ and can be characterized by an effective number of dimensions $m_{eff} \sim \sqrt{N_{nI}}$. In this case, instead of the $\mathcal{O}(m^2)$ scaling with the number of variables, the complexity of parameter inference highly depends on the sparsity of the dependencies in the graphical model and scales with as $\mathcal{O}(n \cdot N_{nI}) \sim \mathcal{O}(n \cdot m_{eff}^2)$.

Therefore, the our method is especially efficient on the datasets with a low effective dimensionality $m_{eff}$, such as the neuronal data. The dimensionality $m$ itself has little effect on the computational cost and memory requirements, as the total complexity scales as $n \cdot (m_{eff}^2 + c \cdot m^2)$, where $c$ is a small constant ($\approx 0.05$) equal to the ratio between independency test time to the extra time required for model selection to complete. However, the second term can become significant for extremely large datasets, where many bivariate models have to be tested for dependence vs. independence. If this becomes an issue, some additional independence assumptions (e.g. for distant neurons) can be incorporated into the model.

## 9 Generation of the synthetic calcium recording from a GLM model

We generated the synthetic spike counts using a Generalized linear model (GLM) with an exponential non-linearity and Poisson emission model [38]:

$$\lambda_i = 0.2 \cdot \exp(\mathbf{1}^{1 \times T} x^{-T:0} + \sum_j h_{ij} y_j^{-T:0}) \tag{6}$$

$$y_i^0 \sim \text{Poisson}(\lambda_i) \tag{7}$$

where $T = 20$ is the length of the filters, $x^{-T:0}$ are the values of the stimuli in the preceding $T$ steps (up to, but excluding 0) of size $(T \times 1)$, $y^{-T:0}$ is a matrix $(T \times 2)$ of neural responses, and $h$ is a tensor $(2 \times 2 \times T)$ of coupling filters:

$$h(t) = \begin{pmatrix} -0.1 & 0.4(-t/T)^2 \\ -0.1(1+t/T)^2 & -0.1 \end{pmatrix}, t \in [-T, 0] \tag{8}$$

for coupled neurons in Fig 3. For independent neurons in Fig 2, the off-diagonal elements were equal to 0.

We simulate 100 identical trials with fixed duration ($\tau = 100$ steps), in which the same sequence of stimuli $x(t)$ was applied (see, for example, Fig 2A). In order to generate continuous

data mimicking the calcium optical recordings, we convolve the simulated spike count data $\mathbf{y}(t)$ with an exponential temporal kernel (as in Deneux et al. [39] model). A characteristic time of the exponent in the synthetic calcium dynamics was 4 time steps (whole interval $t \in [0, \tau]$, $\tau = 100$ steps). The resulting calcium traces are shown in Figs 2B, 3B and 3C.

## 10 Goodness-of-fit

We measure the accuracy of the density estimation with the proportion of variance explained $R^2$. We compare the empirical conditional CDF $\mathrm{ecdf}(u_2|u_1 = y)$ vs. estimated conditional CDF $\mathrm{ccdf}(u_2|u_1 = y)$ and calculate:

$$R^2(y) = 1 - \sum_{u_2} \left( \frac{\mathrm{ecdf}(u_2|u_1 = y) - \mathrm{ccdf}(u_2|u_1 = y)}{\mathrm{ecdf}(u_2|u_1 = y) - \overline{u_2}} \right)^2, \tag{9}$$

where $R^2(y)$ quantifies the portion of the total variance of $u_2$ that our copula model can explain given $u_1 = y$, and $\overline{u_2} = \overline{F(y_2)} = 0.5$. The sum was calculated for $u_2 = 0.05\, n$, $n = 0\ldots20$.

Next, we select all of the samples from a certain interval of the task ($x \in [x_1, x_2]$) matching one of those shown in Fig 3 in the paper. We split these samples $u_1 \in [0, 1]$ into 20 equally sized bins: $\{I_i\}_{20}$. For each bin $I_i$, we calculate (9). We evaluate $\mathrm{ccdf}(u_2|u_1 = y_i) \approx \mathrm{ccdf}(u_2|u_1 \in I_i)$ using a copula model from the center of mass of the considered interval of $x$: $x_\mu = \mathrm{mean}(x)$ for samples $x \in [x_1, x_2]$. We use the average measure:

$$\overline{R^2} = \mathop{\mathbb{E}}_{p(u_1 \in I_i)} R^2(\mathrm{mean}(u_1 \in I_i)), \tag{10}$$

to characterize the goodness of fit for a bivariate copula model. Since $u_1$ is uniformly distributed on $[0, 1]$, the probabilities for each bin $p(u_1 \in I_i)$ are equal to $1/20$, and the resulting measure $\overline{R^2}$ is just an average $R^2$ from all bins. The results were largely insensitive to the number of bins (e.g. 20 vs. 100).

## 11 Entropy and mutual information

Our framework provides tools for efficient sampling from the conditional distribution and for calculating the probability density $p(\mathbf{y}|x)$. Therefore, for each $x = t$ the entropy $H(\mathbf{y}|x = t)$ can be estimated using Monte Carlo (MC) integration: $H(\mathbf{y}|x = t) = -\mathbb{E}_{p(\mathbf{y}|x=t)} \log p(\mathbf{y}|x = t)$. The probability $p(\mathbf{y}|x = t)$ factorizes into the conditional copula density and marginal densities (2), hence the entropy also factorizes [18] as $H(\mathbf{y}|x = t) = \sum H(y_i|x = t) + H_c(\mathbf{u}^x|x = t)$, where $\mathbf{u}^x = \mathbf{F}(\mathbf{y}|x)$. The conditional entropy can be integrated as

$$H(\mathbf{y}|x) = \sum_{i=1}^{N} H(y_i|x) + \int H_c(\mathbf{u}^x|x = t) p(t) dt, \tag{11}$$

separating the entropy of the marginals $\{y_i\}_N$ from the copula entropy.

Now, $I(x, \mathbf{y}) = I(x, \mathbf{G}(\mathbf{y}))$ if $\mathbf{G}(\mathbf{y})$ is 1) a homeomorphism, 2) independent of $x$ [53]. If marginal statistics are independent of $x$, then the probability integral transform $\mathbf{u} = \mathbf{F}(\mathbf{y})$ satisfies both requirements, and $I(x, \mathbf{y}) = I(x, \mathbf{u})$. Then, in order to calculate the mutual information $I(x, \mathbf{u}) := H(\mathbf{u}) - H(\mathbf{u}|x)$, we must also rewrite it using only the conditional distribution $p(\mathbf{u}|x)$, which is modelled with our Copula-GP model. This can be done as follows:

$$I(x, \mathbf{u}) = H(\mathbf{u}) - \int H(\mathbf{u}|x = t) p(t) dt = \mathop{\mathbb{E}}_{p(\mathbf{u}, x)} \log p(\mathbf{u}|x) - \mathop{\mathbb{E}}_{p(\mathbf{u})} \log \mathop{\mathbb{E}}_{p(x)} p(\mathbf{u}|x). \tag{12}$$

The last term in (12) involves nested integration, which is computationally difficult and does not scale well with $N = \dim \mathbf{u}$. Therefore, we propose an alternative way of estimating $I(x, \mathbf{y})$, which avoids double integration and allows us to use the marginals conditioned on $x$ ($\mathbf{u}^x = \mathbf{F}(\mathbf{y}|x)$). We can use two separate copula models, one for estimating $p(\mathbf{y})$ and calculating $H(\mathbf{y})$, and another one for estimating $p(\mathbf{y}|x)$ and calculating $H(\mathbf{y}|x)$:

$$I(x, \mathbf{y}) = \sum_{i=1}^{N} I(x, y_i) + H_c(u_1, \ldots, u_N) - \int H_c(u_1^x, \ldots, u_N^x | s = t) p(t) dt, \tag{13}$$

where both entropy terms are estimated with MC integration. Here we only integrate over the unit cube $[0, 1]^N$ and then parameter domain domx, whereas (12) required integration over $[0, 1]^N \times \dim x$.

The performance of both (12) and (13) critically depends on the approximation of the dependence structure, i.e. how well the parametric copula approximates the true copula probability density. If the joint distribution $p(y_1 \ldots y_N)$ has a complex dependence structure, as we saw in synthetic examples (Sec. 3), then the mixture of parametric copulas may provide a poor approximation of $p(\mathbf{y})$ and overestimate $H_c(u_1, \ldots, u_N)$, thereby overestimating $I(x, \mathbf{y})$. The direct integration (12), on the other hand, typically underestimates the $I(x, \mathbf{y})$ due to imperfect approximation of $p(\mathbf{y}|x)$, and is only valid under assumption that the marginals can be considered independent of $x$.

We refer to the direct integration approach (12) as "Copula-GP integrated" and to the alternative approach (13) as "Copula-GP estimated" and assess both of them on synthetic and real data.

## Supporting information

**S1 Text. Supplementary methods.** Include Bayesian model selection details and mixture model construction strategies.
(PDF)

**S2 Text. Supplementary model identifiability tests.** Include the description of synthetic data generation and the results of mixture model identifiability tests.
(PDF)

**S3 Text. Supplementary model validation.** Describes validation of entropy estimation for Gaussian and Clayton copulas, validation on UCI benchmarks, comparison between single copula and mixture models, and additional tests that explore limitations of empirical marginal estimation.
(PDF)

**S4 Text. Model parameters interpretation.** Includes additional parameter visializations and an ablation study for the copulas in the mixtures shown in Fig 5.
(PDF)

## Acknowledgments

We thank the GENIE Program and the Janelia Research Campus, specifically V. Jayaraman, R. Kerr, D. Kim, L. Looger, and K. Svoboda, for making GCaMP6 available.

## Author Contributions

**Conceptualization:** Nathalie Rochefort, Arno Onken.

**Data curation:** Nina Kudryashova, Theoklitos Amvrosiadis, Nathalie Dupuy.

**Formal analysis:** Nina Kudryashova.

**Funding acquisition:** Nathalie Rochefort, Arno Onken.

**Investigation:** Nina Kudryashova, Theoklitos Amvrosiadis.

**Methodology:** Nina Kudryashova, Arno Onken.

**Project administration:** Arno Onken.

**Resources:** Nathalie Rochefort.

**Software:** Nina Kudryashova.

**Supervision:** Nathalie Rochefort, Arno Onken.

**Validation:** Nina Kudryashova.

**Visualization:** Nina Kudryashova.

**Writing – original draft:** Nina Kudryashova, Arno Onken.

**Writing – review & editing:** Nina Kudryashova, Theoklitos Amvrosiadis, Nathalie Dupuy, Nathalie Rochefort, Arno Onken.

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
