## [Decision Letter · Decision Letter 0]

18 Oct 2021

Dear Dr Kudryashova,

Thank you very much for submitting your manuscript "Parametric Copula-GP model for analyzing multidimensional neuronal and behavioral relationships" for consideration at PLOS Computational Biology.

As with all papers reviewed by the journal, your manuscript was reviewed by members of the editorial board and by several independent reviewers. In light of the reviews (below this email), we would like to invite the resubmission of a significantly-revised version that takes into account the reviewers' comments.

I would like to apologise for the length of time the review process has taken for this manuscript. Unfortunately there was a reviewer who did not respond and had to withdraw from the process.

Please find attached the three reviewers reports. I would like to invite you to make revisions to address the reviewers comments.

We cannot make any decision about publication until we have seen the revised manuscript and your response to the reviewers' comments. Your revised manuscript is also likely to be sent to reviewers for further evaluation.

Sincerely,

Robin A A Ince, Ph. D.

Guest Editor

PLOS Computational Biology

Thomas Serre

Deputy Editor

PLOS Computational Biology

I would like to apologise for the length of time the review process has taken for this manuscript. Unfortunately there was a reviewer who did not respond and had to withdraw from the process.

Please find attached the three reviewers reports. I would like to invite you to make revisions to address the reviewers comments.

Reviewer's Responses to Questions

**Comments to the Authors:**

Reviewer #1: The manuscript develops the usage of a combination of copulas and Gaussian processes to capture interdependencies in diverse scenarios. The idea is interesting and could be potentially very useful. Additionally, the paper is well written and the technical aspects seem solid. However, I had a hard time to judge the value of the actual contribution, for reasons I explain below.

First, the actual contribution of the paper is not clear. The abstract and introduction implicitly suggest that the idea of combining copulas and Gaussian processes is an original idea of the authors, but then in the result sections it is clarified that actually this has already been developed before in the context of econometric time series data. So, the novelty of the paper seems to be a set of adjustments that allow the translation of these techniques to become suitable to deal with specific types of neural data, but this is not entirely clear. The authors should clarify what are exactly the technical contributions of the paper.

As a second point, the paper is focused in a new method to estimate interdependencies, but most of the paper is devoted to the development of examples and little new formal analyses are provided. While the detailed development of these examples may be appreciated by the specialist, in my opinion they may not suffice to establish general & rigorous guarantees that could guide the general usage of these techniques. The authors should complement these examples with more formal analyses, or at least provide strong arguments of why the presented examples are enough to assume that what they illustrate should generalise.

Reviewer #2: With the increasing complexity of systems neuroscience datasets (both in terms of the number

of simultaneously recorded neurons, and in terms of increasing task complexity) there is a

pressing need for large scale statistical models that are flexible, yet can be estimated from

relatively limited measurements. The copula family of models (which separately models

marginal statistics and inter-variable interactions) provides a promising but still relatively

unexplored framework for constructing such models.

The manuscript integrates copulas with complex structure (copula mixtures for pairwise

interactions, smooth dependencies over time in the parameters, and vine copulas for joint

statistics) to construct statistical models of stimulus-/time-varying network interactions.

Moreover, behavioral variables can be integrated as extra dimensions in the joint model.

This framework is applied on simulated data, where it shows sensible reconstruction of

underlying data statistics and improved accuracy in mutual information estimates based on the

fitted model compared to other state-of-the-art methods. The model is also fitted to a

neuroscience dataset from rodents navigating in a VR linear track in order to get a reward.

Fitting the model on this data reveals sparse time-varying noise correlations, which seem to

reflect the constraints of the task and the low dimensional structure of the associated neural

responses.

The paper is technically very accomplished, as it brings a range of state of the art machine

learning tricks into the statistical toolkit of neuroscience. However, although the presentation of

the results generally clear, it does demand a fairly large amount of technical background

knowledge from the reader; moreover, I found that the text spends too little time exploring the

neuroscientific relevance of the information extracted via the new statistical tool. There are

some missed opportunities in engaging with the broader systems neuroscience community,

both in terms of explaining what the model is and in terms of educating the reader about how

the statistical features extracted by the model can be translated back into neuroscientific

knowledge.

Some comments and/or suggestions for text improvement :

(the order roughly follows that in which the information appears in the text)

1. Some more layman introduction into copulas at the beginning of the results could help

define the model not just as a mathematical construct but also in analogy with the real

world statistical features that a neuroscientist would care about. Motivate the model by the

kind of statistical questions that a neuroscientists want to answer — whenever possible.

2. Line 72: there is an emphasis that the conditioning variable needs to be continuous that I

find a bit misleading since KISS-GP and co discretize the variable range

3. Also: I wonder if it would be useful to add a subscript t for conditioning variables that are

time-dependent (within a trial); may help clean up the conditioning on t vs conditioning on x

discussion, which I found could have been explained in a cleaner way

4. Line 100: ‘spike inference algorithm’ are usually referred to as ‘deconvolution’ algorithm,

here I’d add a concrete reference, for instance the Paninski or similar

5. More general in Results section 2: the assumption that two independent neurons should

result into an independent copula, when applied to simulated calcium imaging traces

implicitly assumes a factorized ‘measurement’ model, which it’s not clear to me always

applies to real calcium imaging data. More generally, the very practically relevant distinction

of statistically modeling neural activity directly vs an indirect variable that is correlated to

the quantities of interest, here calcium imaging measurements, is swept under the rug a bit

too much for my liking.

6. Line 114: There is a bit of confusion in the description of the statistical treatment of the

marginals. It is advertised as something that ensures maximal flexibility, but not sure that is

strictly true: the kernel density estimator makes some simplifying assumptions to get a

density from limited data;

7. Related: how much data is needed to get the empirical cdf well estimated without any

assumptions? This seems particularly important when estimating conditional marginals,

since the number of samples is given by the number of repeats of the stimulus, which tends

to be pretty restricted in practice…

8. For the general reader it may important to articulate the distinction between measuring

instantaneous dependencies that are time varying vs time-lagged interactions of the kind

that a GLM temporal filter would capture. Also there seem to be implicit assumptions built

in about the time scale of dependencies that need modeling, which may be helpful to spell

out more directly; the coupling terms in a GLM typically imply synaptic interactions

between spikes at a ms time scale. What is the time scale of interactions that can be

modeled in the copula framework in comparison?

9. A comment on the general philosophy of the toy examples as sanity checks that the

estimator behaves correctly: in ML one would typically test within model parameter

reconstruction first, but this is not quite what is going on here. While the data is in some

sense within the copula model class, the parametrization is very different so it is not at all

immediately clear what should one predict about the shape of the extracted copulas in

each scenario (especially when the neural responses pass through a not completely

specified imaging measurement model )

10. I did not completely understand the depiction of the GLM filters in fig2A, is there a

biophysical rationale for that particular choice (more text about the setup of that toy data

example would be generally helpful).

11. Missed opportunity: educate the reader about what the shape of a copula tells you about

the nature of the interactions between the pair of neurons; the goal is to make those 2d

plots intuitive to decipher for a neuroscientist, ideally. Line 160: this dependent is nonlinear

comes out of nowhere

12. Line 171: I am curious about the technical process that allows one to propagate uncertainty

in copula parameters into decoding/MI estimates: is that something one can do in

analytically or does it rely on sampling from the approx posterior?

13. Line 179: is the fact that the noise correlations are tuned to a particular time point

surprising or a ground truth confirmation type of thing? Don't get where that comes from

(you only mention something about asymmetric connections after the fact; narratively,

would have been better to properly describe the ‘ground truth’ spell out predictions about

what should an estimator find and then numerically confirm that it comes out as expected).

14. Line 202: would have liked a bit more technical detail on the distinction between the two

estimators and why one needs to fit an additional copula model only for one of them.

15. In general: MI is one potential quantity of interest but not necessarily the one that most

would aim to compute when fitting a joint statistical model to neural responses. One could

do a better job in the intro managing expectations and justifying why the main measure of

success in this paper is the ability to estimate reasonable (If still biased) MIs.

16. More general: it is not clear to me how should an experimentalist looking at a new dataset

set out to design hir copula mixture components, especially since the results seem to

suggest that it’s really important to get that right (cf line 289)

17. In the real data: I got really confused about the map between t and x (since self-paced the

speed is variable within and across trials so the two are not interchangeable); as far as I

understand the conditioning variable ended up being location on track, at a relatively

coarse discretization

18. I did not understand what exactly were the behavioral variables that were being jointly fitted

with the neural responses, the phrasing in line 347 only added to the confusion about what

is being conditioned on and what is being jointly fitted.

19. I was struggling to find the take home message for the probability density plots in fig 5CD:

again, i see challenges in interpretation for these observations. what does it mean for the

neural coding properties of the circuit, and is this something expected or independently

verified by some other means? Is there a simpler model that would already capture that

kind of structure?

20. I also did not understand what were the variables between which MI was computed in line

407, which made the increase of MI in the reward zone also difficult to map into neural

coding properties of potential interest

21. It would help if you could add intuitive explanation of how a vine copula works and what

those parameters map into, at the minimal level required to understand the results in sec

4.3; this is just a more expanded vs of the question what does jointly modeling the

behavioral variables teaches us abut the coding properties of the recorded neurons

22. Maybe I missed it but the comparison to other methods makes statements about sample

complexity of estimator (how much data needed, e.g. line 481) that I did not see any direct

support in the results for

23. The statement that other joint models cannot do conditionals, stimulus dependencies etc is

simply not true. There exists at least one stimulus dependent ising model (from Tkacik and

co), and many latent dynamical systems models of joint activity include explicit stimulus

dependencies either in the observation model (e.g. PLDS by Macke) or in the latent space

(e.g. recent work from Scott Linderman)

24. Methods: automatizing hyperparameter optimization seems rather important to make the

tool usable by other on new datasets

25. Methods: can one and maybe should one regularize the components of the mixture in

some way (thinking sparse mixtures may be a bit more interpretable, but then maybe not)

26. Big picture: it is at the very list unusual when modeling neural activity simultaneously with

stimuli and/or behavior that one conditions on the stimulus, but not the behavioral outcome

(theoretical frameworks usually talk about encoding or decoding models of circuit

computation, and this is neither). I think more justification is necessary about the practical

use of such a joint model in interpreting what the circuit computes in relation to the

stimulus and behavior.

27. Ultimately, having a great statistical model of the data is not all that counts when it comes

to basic science; some may argue (and have argued) that a simpler interpretable model

that captures reasonably the data is in some sense better than a complex yet

uninterpretable model with a better quantitative fit. For the tool to be widely used, it needs

not only to do well in ML terms but to extract scientific knowledge — this is the point were

the argument fails short as presented at the moment. At the very least, one should provide

clearer guidelines on what kind of questions is the framework more or less suitable for,

within the space of joint statistical descriptions of neural responses during behavior.

Overall, I think this is a pretty good paper already and my criticism is meant to make it better

by increasing the long term impact of the tool on the community. Not everything needs to

necessarily be addressed for the manuscript to be worthy of publication, but I do think

addressing as many as possible will make the paper ultimately stronger.

Reviewer #3: the review is uploaded as an attachment

**Have the authors made all data and (if applicable) computational code underlying the findings in their manuscript fully available?**

Reviewer #1: Yes

Reviewer #2: Yes

Reviewer #3: Yes

PLOS authors have the option to publish the peer review history of their article (what does this mean?). If published, this will include your full peer review and any attached files.

Reviewer #1: No

Reviewer #2: No

Reviewer #3: **Yes: **Giovanni Diana
---

## [Decision Letter · Decision Letter 1]

2 Jan 2022

Dear Dr Kudryashova,

We are pleased to inform you that your manuscript 'Parametric Copula-GP model for analyzing multidimensional neuronal and behavioral relationships' has been provisionally accepted for publication in PLOS Computational Biology.

Best regards,

Robin A A Ince, Ph. D.

Guest Editor

PLOS Computational Biology

Thomas Serre

Deputy Editor

PLOS Computational Biology

Reviewer's Responses to Questions

**Comments to the Authors:**

Reviewer #1: I congratulate the authors for a very thoughtful revision. I think the manuscript is now in good shape for publication.

Reviewer #2: the authors have reasonably addressed my original concerns

Reviewer #3: In the revised manuscript the authors have substantially improved the clarity of the presentation of the new method and the background on copula technique. The authors have fully addressed all my comments, therefore I am happy to recommend their revised manuscript for publication.

**Have the authors made all data and (if applicable) computational code underlying the findings in their manuscript fully available?**

Reviewer #1: None

Reviewer #2: Yes

Reviewer #3: Yes

PLOS authors have the option to publish the peer review history of their article (what does this mean?). If published, this will include your full peer review and any attached files.

Reviewer #1: No

Reviewer #2: No

Reviewer #3: No

---

## [Editor Report · Acceptance letter]

19 Jan 2022

PCOMPBIOL-D-21-01071R1 

Parametric Copula-GP model for analyzing multidimensional neuronal and behavioral relationships

Dear Dr Kudryashova,

I am pleased to inform you that your manuscript has been formally accepted for publication in PLOS Computational Biology. Your manuscript is now with our production department and you will be notified of the publication date in due course.

With kind regards,

Anita Estes
